# MITOL prevents ER stress-induced apoptosis by IRE1α ubiquitylation at ER–mitochondria contact sites

Keisuke Takeda[1], Shun Nagashima[1], Isshin Shiiba[1], Aoi Uda[1], Takeshi Tokuyama[1], Naoki Ito[1], Toshifumi Fukuda[1], Nobuko Matsushita[1] (iD), Satoshi Ishido[2], Takao Iwawaki[3], Takashi Uehara[4], Ryoko Inatome[1] & Shigeru Yanagi[1,*] (iD)

## Abstract

Unresolved endoplasmic reticulum (ER) stress shifts the unfolded protein response signaling from cell survival to cell death, although the switching mechanism remains unclear. Here, we report that mitochondrial ubiquitin ligase (MITOL/MARCH5) inhibits ER stress-induced apoptosis through ubiquitylation of IRE1α at the mitochondria-associated ER membrane (MAM). MITOL promotes K63-linked chain ubiquitination of IRE1α at lysine 481 (K481), thereby preventing hyper-oligomerization of IRE1α and regulated IRE1α-dependent decay (RIDD). Therefore, under ER stress, MITOL depletion or the IRE1α mutant (K481R) allows for IRE1α hyper-oligomerization and enhances RIDD activity, resulting in apoptosis. Similarly, in the spinal cord of MITOL-deficient mice, ER stress enhances RIDD activity and subsequent apoptosis. Notably, unresolved ER stress attenuates IRE1α ubiquitylation, suggesting that this directs the apoptotic switch of IRE1α signaling. Our findings suggest that mitochondria regulate cell fate under ER stress through IRE1α ubiquitylation by MITOL at the MAM.

**Keywords** apoptosis; IRE1α; mitochondria-associated ER membrane; mitochondrial E3 ligase MITOL/MARCH5; unfolded protein response
**Subject Categories** Autophagy & Cell Death; Membrane & Intracellular Transport; Post-translational Modifications, Proteolysis & Proteomics
**The EMBO Journal (2019) 38: e100999**

## Introduction

The endoplasmic reticulum (ER) is an intracellular compartment essential for the maturation of proteins. Various physiological and pathological changes require enhanced ER functions, such as protein folding, removal of unfolded proteins, and reduction of oxidative stress. An imbalance in ER homeostasis causes ER stress and triggers the unfolded protein response (UPR), which is initiated by three ER proteins, PERK, ATF6, and IRE1α. The UPR is involved in the recovery of ER homeostasis through ERAD and ER chaperones (Travers *et al*, 2000). However, under persistent or severe ER stress, an alternative UPR triggers apoptosis (Shore *et al*, 2011; Tabas & Ron, 2011). UPR-mediated apoptosis is responsible for the pathogenesis of human diseases including diabetes mellitus, heart failure, and neurodegenerative diseases (Yoshida, 2007). Therefore, understanding the mechanisms of UPR regulation may contribute to the development of therapies for these diseases.

IRE1α is the most highly conserved UPR sensor protein possessing bifunctional activities, kinase, and RNase (Wang *et al*, 1998). Upon ER stress, IRE1α directly or indirectly recognizes unfolded proteins, leading to the autophosphorylation and oligomerization of IRE1α (Bertolotti *et al*, 2000; Korennykh *et al*, 2009). The oligomerization of IRE1α is required for its RNase activity (Gardner & Walter, 2011). IRE1α splices *xbp1* mRNA to generate the mature protein XBP1s, which induces the expression of a group of genes involved in the quality control of the ER (Cox & Walter, 1996). However, when ER stress is severe or unresolved, IRE1α cleaves various mRNAs localized in the ER, which are mainly dependent on translational attenuation mediated by PERK (Moore & Hollien, 2015). This mRNA decay mediated by IRE1α is known as regulated IRE1α-dependent decay (RIDD) (Hollien *et al*, 2009). Anti-apoptotic miRNA is also degraded by IRE1α, resulting in increased mRNA and protein expression of TXNIP and caspase-2 (Lerner *et al*, 2012; Upton *et al*, 2012). In contrast, IRE1α phosphorylates ASK1 and JNK to trigger the intrinsic apoptotic pathway through mitochondrial depolarization (Urano *et al*, 2000; Nishitoh *et al*, 2002). It has been reported that excessive oligomerization of IRE1α leads to apoptotic signaling induced by enhanced IRE1α RNase and kinase activities (Ghosh *et al*, 2014); however, the regulatory mechanisms

1   Laboratory of Molecular Biochemistry, School of Life Sciences, Tokyo University of Pharmacy and Life Sciences, Hachioji, Tokyo, Japan
2   Department of Microbiology, Hyogo College of Medicine, Nishinomiya, Japan
3   Medical Research Institute, Kanazawa Medical University, Ishikawa, Japan
4   Department of Medicinal Pharmacology, Graduate School of Medicine, Dentistry, and Pharmaceutical Sciences, Okayama University, Okayama, Japan
    *Corresponding author. Tel: +81 42 676 7146; E-mail: syanagi@ls.toyaku.ac.jp
    [The copyright line of this article was changed on 26 June 2019 after original online publication.]

underlying IRE1α hyper-oligomerization and the apoptotic switch of IRE1α signaling remain largely unknown.

The unique region of the ER membrane that connects to the mitochondria is denoted as the mitochondria-associated ER membrane (MAM) and plays a central role in the lipid metabolism and Ca$^{2+}$ transport between the ER and mitochondria (Raturi & Simmen, 2013). A recent study showed that PERK is mainly localized at the MAM and transmits apoptotic signals from the ER to mitochondria (Verfaillie *et al*, 2012). In contrast, IRE1α is also known to be enriched at the MAM to promote cell survival by splicing the *xbp1* mRNA (Mori *et al*, 2013); conversely, IRE1α at the MAM can also induce cell death by mitochondrial Ca$^{2+}$ overload (Son *et al*, 2014). These findings indicate that determining the role of MAM in UPR regulation is an urgent issue.

We have previously identified mitochondrial ubiquitin ligase (MITOL/MARCH5) and demonstrated that MITOL ubiquitylates mitofusin 2 (Mfn2) and enhances its GTPase activity, resulting in the tethering between the ER and mitochondria rather than mitochondrial fusion (Sugiura *et al*, 2013). Notably, the silencing of MITOL expression was found to reduce the integrity of the ER network, which may be caused not only by Mfn2 inactivation but also by an unknown mechanism. Several studies have suggested that MITOL contributes to the regulation of mitochondrial dynamics and various signaling pathways (Nagashima *et al*, 2014; Yoo *et al*, 2015; Cherok *et al*, 2017). However, the functional relationship between MITOL and the ER remains to be fully elucidated.

In this study, we identified IRE1α as a novel substrate for MITOL. MITOL modifies the lysine (K) 481 of IRE1α by adding a K63-linked polyubiquitin chain, which in turn suppresses the activity and oligomerization of IRE1α. Here, we provide evidence that MITOL can directly regulate IRE1α through the MAM and thus play an important role in determining cell fate under ER stress.

## Results

### MITOL plays an anti-apoptotic role against ER stress

In a previous study, we generated 4-hydroxytamoxifen (4-OHT)-inducible MITOL-knockout MEFs by flanking two loxP sequences with exon 2 of the *mitol/march5* gene (Sugiura *et al*, 2013). Here, we classified cells into two types: normal or abnormal morphologies of mitochondria or the ER (Fig EV1A and B). Acute depletion of MITOL by 4-OHT treatment caused morphological abnormalities not only in the mitochondria, but also in the ER (Fig EV1C and D). Since an imbalance in mitochondrial homeostasis has been shown to cause ER stress, we suspected that the morphological abnormalities in the ER of 4-OHT-treated MITOL$^{F/F}$ MEFs could have resulted from mitochondrial fragmentation. Drp1 knockdown by shDrp1#1 and shDrp1#2 rescued the morphological abnormalities in the mitochondria of MITOL$^{F/F}$ MEFs treated with 4-OHT, whereas Drp1 silencing did not affect the morphological abnormalities in the ER of these cells (Fig EV1E–G), suggesting that the role of MITOL in the maintenance of ER homeostasis is independent of its function in mitochondrial dynamics via Drp1 regulation.

To characterize the role of MITOL in the maintenance of ER homeostasis, we used MEFs treated with 4-OHT at least 2 weeks prior to the experiment. Compared with acute MITOL depletion, MEFs under chronic MITOL knockout (KO) showed abnormal morphology of the ER but not of the mitochondria (Fig EV1H and I). Moreover, the amount of resting Ca$^{2+}$ in the ER was increased in MITOL-KO MEFs (Fig EV1J). We first used MITOL-KO MEFs to investigate the effects of three ER stress inducers, thapsigargin, tunicamycin, and brefeldin A. MITOL depletion dramatically reduced the survival rates of MEFs under ER stress (Fig 1A). MITOL-KO MEFs exhibited a high level of phosphatidylserine exposure under ER stress (Fig 1B). To further demonstrate the anti-apoptotic role of MITOL under ER stress, we performed rescue experiments with re-expression of wild-type MITOL or MITOL CS mutant lacking enzymatic activity (C65/68S). Tunicamycin-treated MITOL-KO MEFs showed increased cleavage of caspase-3 and PARP; however, these activations were recovered upon re-expression of MITOL (Fig 1C). In contrast, the re-expression of the MITOL CS mutant did not rescue the cleavage of caspase-3 and PARP in MITOL-KO MEFs (Fig 1C). These results show that the enzymatic activity of MITOL contributes to cell survival under ER stress.

ER stress has been shown to trigger apoptosis via mitochondrial permeability transition pore (Reimertz *et al*, 2003). To investigate the mechanisms by which MITOL determines cell fate under ER stress, we monitored the activation of BAX and the release of cytochrome c, key regulators of the mitochondrial permeability transition pore, and subsequent apoptosis. In response to tunicamycin treatment, MITOL depletion enhanced mitochondrial recruitment of BAX and subsequently promoted the release of cytochrome c (Fig 1D). A drastic mitochondrial depolarization was also observed in MITOL-KO MEFs treated with tunicamycin (Fig 1E). Mitochondrial depolarization in MITOL-KO MEF was restored by re-expressing wild-type MITOL, but not the MITOL CS mutant (Fig 1E). These findings suggest that, under ER stress, MITOL exerts anti-apoptotic function upstream of mitochondrial depolarization.

### MITOL blocks IRE1α-mediated mRNA decay and JNK phosphorylation

Three UPR sensor proteins, PERK, ATF6, and IRE1α, induce opposing UPR signals, which mediate either the adaptation pathway by *xbp1* splicing and ATF6 cleavage or the cell death pathway by enhanced RIDD activity and CHOP expression (Shore *et al*, 2011). We evaluated the activities of the UPR sensor proteins. MITOL depletion did not affect autophosphorylation of PERK, induction of the downstream genes *chop* and *atf4*, or ATF6 cleavage (Fig EV2A–C). Moreover, MITOL was dispensable for gene expression of DR5 under ER stress (Fig EV2D). Therefore, we hypothesized that MITOL regulates ER stress-induced apoptosis via IRE1α. To confirm that IRE1α promotes apoptosis in MITOL-KO MEFs, we investigated the effect of IRE1α inhibition. Cleavage of caspase-3 in MITOL-KO MEFs was attenuated by siIRE1α (Fig 2A) or treatment with 4μ8C, an inhibitor of IRE1α RNase (Fig 2B), suggesting that MITOL inhibits ER stress-dependent cell death via IRE1α. Given that MITOL did not significantly affect IRE1α autophosphorylation under ER stress (Fig 2C), we examined two opposing signals induced by IRE1α, adaptation, and cell death. Although tunicamycin-treated MITOL-KO MEFs displayed significant cell death (Fig 1), MITOL-KO MEFs exhibited an increase in *xbp1* splicing, which is an IRE1α-dependent adaptation response, and expression of the downstream genes *edem*, *sec61*, and *herp* (Fig 2D). Notably, MITOL depletion robustly

**A**

**B**

**C**

**D**

**E**

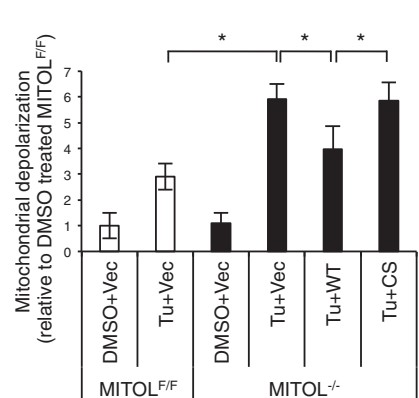

**Figure 1.**

**Figure 1. Anti-apoptotic role of MITOL against ER stress.**

A   MITOL-KO MEFs were vulnerable to ER stress. Control MEFs (MITOL$^{F/F}$) and MITOL-KO MEFs (MITOL$^{-/-}$) were treated with DMSO as control, 0.8 μM thapsigargin (Tg), 0.7 μg/ml tunicamycin (Tu), or 1.2 μg/ml brefeldin A (Br) for 24 h. All experiments using Tu below were performed at concentration of 0.7 μg/ml. The relative cell viability shows a ratio between the survival of MEFs in the absence and presence of ER stress inducers. Viable cells were detected by cell viability assay using Cell Counting Kit-8. Error bars represent SD ($n = 6$). $n$ is the number of independent experiments. ***$P < 0.001$ (Student's $t$-test).

B   MITOL KO increased ER stress-induced apoptosis. MEFs were treated with indicated ER stress inducers at same concentration as Fig 1A for 18 h and stained with Annexin V-FITC for the detection of apoptosis using flow cytometry. Error bars represent SD ($n = 3$). *$P < 0.05$ (Student's $t$-test).

C   MITOL re-expression in MITOL-KO MEFs rescued ER stress vulnerability. MEFs were transfected with MITOL-coding vector (WT), MITOL C65/68S-coding vector (CS), or empty vector (Vec) 24 h prior to Tu treatment for 18 h, followed by immunoblotting with indicated antibodies. Cleaved caspase-3 (cC3) and cleaved PARP (cPARP) were used as apoptosis markers. Error bars represent ± SD ($n = 3$). **$P < 0.01$, ***$P < 0.001$ (Student's $t$-test).

D   MITOL KO enhanced ER stress-induced Bax translocation and cytochrome c release. Mitochondria or cytosol fraction was isolated from MEFs treated with Tu for 18 h, followed by immunoblotting with indicated antibodies. Error bars represent SD ($n = 3$). **$P < 0.01$, ***$P < 0.001$ (Student's $t$-test).

E   MITOL KO induced excessive mitochondrial depolarization by ER stress. MEFs transfected with indicated vectors were treated with Tu for 18 h and then stained with TMRM, followed by flow cytometric analysis. Error bars represent SD ($n = 3$). *$P < 0.05$ (Student's $t$-test).

reduced mRNA expression of RIDD-target genes after tunicamycin treatment (Fig 2E). It is known that IRE1α degrades not only mRNA but also anti-apoptotic miRNA, which in turn leads to the upregulation of TXNIP expression and subsequent apoptosis (Lerner et al, 2012). Therefore, the amount of anti-apoptotic miRNAs, miR-17 and miR-34, was evaluated using a modified luciferase reporter system. In this system, the luciferase reporter construct contained the miRNA binding sites in its 3′UTR; thus, luciferase expression was suppressed by mRNA degradation in the presence of specific miRNAs binding to the 3′UTR of luciferase mRNA. The luciferase reporter assay demonstrated that MITOL deletion significantly reduced both miR-17 and miR-34 following tunicamycin treatment (Fig 2F). Consistently, MITOL-KO MEFs showed higher levels of *txnip* and *caspase-2* mRNAs, the endogenous targets of miR-17 and miR-34, upon tunicamycin treatment (Fig 2G). Considering that IRE1α kinase induces apoptosis through phosphorylation of ASK1 and JNK (Urano et al, 2000; Nishitoh et al, 2002), we next examined tunicamycin-dependent JNK phosphorylation. In MITOL-KO MEFs, JNK phosphorylation was prolonged after tunicamycin treatment (Fig 2H). Indeed, the cleavage of caspase-3 in MITOL-KO MEFs was attenuated by both treatment with a JNK inhibitor or JNK silencing (Figs 2I and EV2E). Taken together, these findings demonstrate that MITOL inhibits ER stress-dependent cell death via the suppression of IRE1α-mediated RIDD activity and JNK phosphorylation.

## MITOL suppresses IRE1α hyper-oligomerization partly via dissociation of BIM

A previous report suggested that MITOL depletion sensitizes cells to various stress conditions dependent on mitochondrial fission induced by the Drp1 and Drp1 receptor MiD49 (Xu et al, 2016). We therefore assessed the effect of Drp1-dependent mitochondrial fission on IRE1α activity in MITOL-KO MEFs. Although Drp1 knockdown strongly suppressed the morphological abnormalities of mitochondria in MITOL-KO MEFs (Fig EV2F and G), shDrp1 did not affect IRE1α activity (Fig EV2H–J), suggesting that MITOL regulates IRE1α activity independent of its function in Drp1-dependent mitochondrial fission. Oligomerization of IRE1α is crucial for activating its RNase activity (Gardner & Walter, 2011). In particular, excessive oligomerization and activation of IRE1α lead to RIDD-mediated cell death (Ghosh et al, 2014). To clarify the mechanism underlying the enhancement of RIDD activity by MITOL depletion, IRE1α

oligomerization was visualized using IRE1α-GFP fusion protein, in which green fluorescent protein (GFP) was inserted between the transmembrane and kinase region of IRE1α (Li et al, 2010). In the control MEFs, IRE1α-GFP was observed as foci-like structures at 4 h following tunicamycin treatment; however, in MITOL-KO MEFs, formation of the foci-like structures by IRE1α-GFP was more efficient and prolonged (Fig 3A). MITOL depletion also promoted endogenous IRE1α hyper-oligomerization (Fig 3B). In MITOL-KO MEFs treated with tunicamycin, the foci-like structures of IRE1α-GFP were maintained long after tunicamycin was removed (Fig 3C). Similarly, autophosphorylation of IRE1α was also sustained by MITOL depletion after tunicamycin was washed out (Fig 3D). IRE1α hyper-oligomerization has been previously shown to trigger cell death rather than cellular adaptation (Ghosh et al, 2014); therefore, MITOL is considered to prevent IRE1α-mediated apoptosis by blocking IRE1α hyper-oligomerization. Phosphorylated PERK and cleaved ATF6 in MITOL-KO MEFs were reduced to level similar to those in the control MEFs after tunicamycin was washed out (Fig EV2K and L). Recent studies have shown that the IRE1α oligomers are stabilized by a protein complex containing IRE1α, denoted as the UPRosome (Woehlbier & Hetz, 2011). BIM is one of the constituent molecules of the UPRosome, and the interaction between IRE1α and BIM stabilizes the IRE1α oligomers (Rodriguez et al, 2012). Interestingly, MITOL depletion increased the association between IRE1α and BIM (Fig 3E). These data indicate that MITOL inhibits IRE1α hyper-oligomerization and RIDD activity at least in part via the regulation of BIM.

## IRE1α is a novel substrate for MITOL

In HEK293 cells, endogenous IRE1α was found to be co-precipitated with endogenous MITOL (Fig 4A). The results of immunoprecipitation assays using HEK293 cells transfected with IRE1α-FLAG and MITOL-HA suggested that MITOL binds to IRE1α (Fig 4B). A GST pull-down assay using GST-fused fragments of either MITOL or IRE1α indicated that the C-terminus of IRE1α directly interacts with the N-terminus of MITOL (Fig 4C and D). We next investigated IRE1α ubiquitylation by MITOL. An immunoprecipitation assay confirmed endogenous IRE1α ubiquitylation by MITOL (Fig 4E). Moreover, the overexpression of MITOL increased IRE1α ubiquitylation compared to that of an empty vector or inactive MITOL mutant (Fig 4F). IRE1α ubiquitylation was inhibited in MITOL-KO MEFs and was rescued only by expression of wild-type MITOL but not by

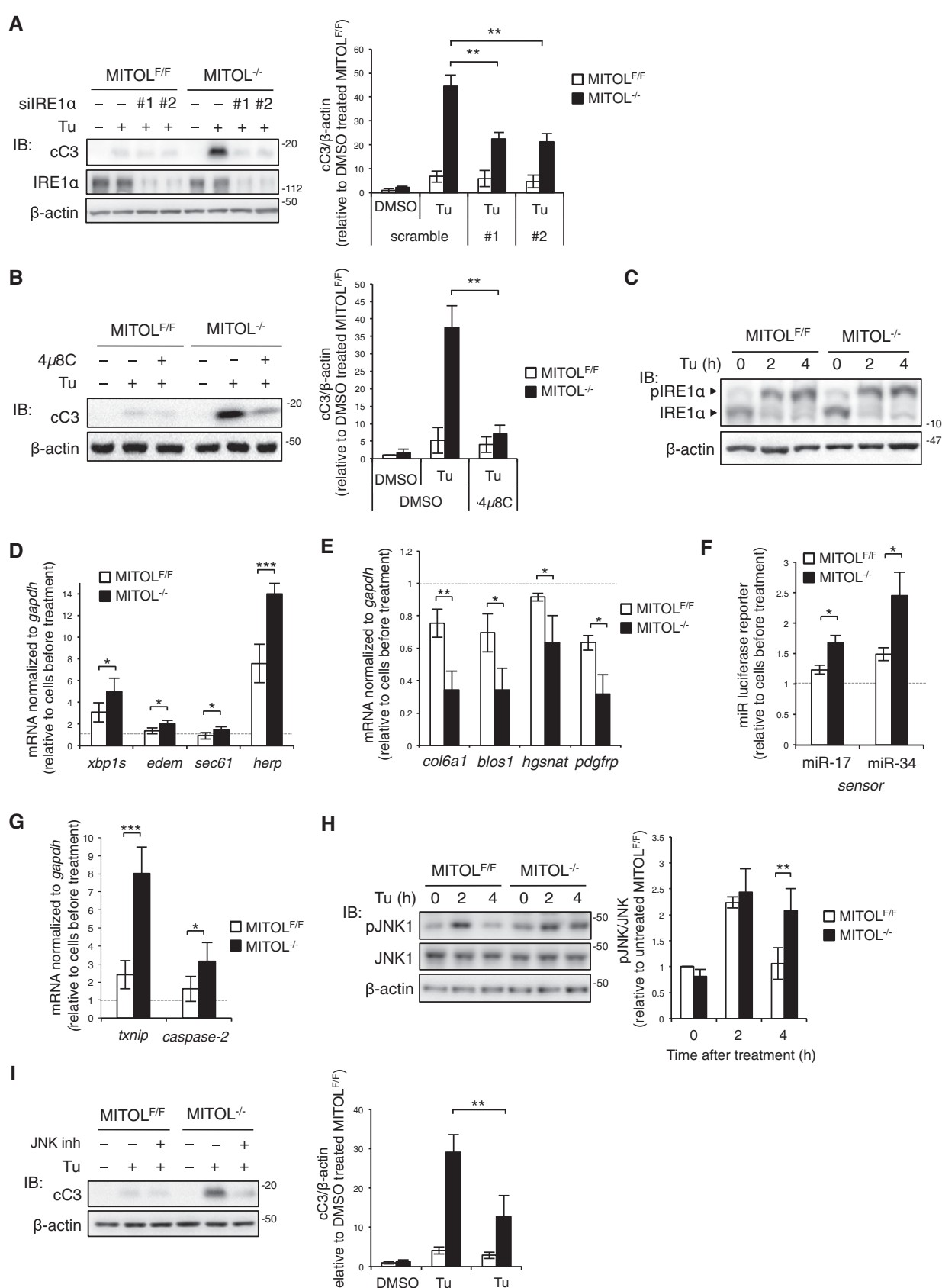

Figure 2.

**Figure 2.  MITOL KO enhances IRE1α-dependent apoptosis under ER stress.**

A, B   IRE1α inhibition compensated MITOL-KO MEFs for vulnerability to ER stress. MEFs were transfected with scramble or siIRE1α (#1, #2). After 24 h, these MEFs were
         exposed with Tu for 24 h (A). MEFs were treated with 4μ8C (25 μM) for 2 h prior to Tu treatment for 24 h (B), followed by immunoblotting with indicated
         antibodies. Error bars represent SD ($n = 3$). **$P < 0.01$ (Student's $t$-test).

C       MITOL KO did not affect the autophosphorylation of IRE1α. MEFs were exposed with Tu for indicated periods, and phosphorylated IRE1α was detected by
         immunoblotting after Phos-tag SDS–PAGE.

D–G   MITOL KO enhanced RIDD and *xbp1* splicing. MEFs were incubated with Tu for 4 h. Expression levels of various UPR-associated genes were determined by qRT–
         PCR (D, E, G). MEFs were transfected with either miR-17 or miR-34 luciferase reporter vector 24 h prior to Tu treatment for 4 h, followed by luciferase reporter
         assay (F). Error bars represent SD ($n = 3$). *$P < 0.05$, **$P < 0.01$, ***$P < 0.001$ (Student's $t$-test).

H       IRE1α-mediated JNK phosphorylation was prolonged by MITOL KO. MEFs were incubated with Tu for indicated periods, and phosphorylated JNK was detected by
         immunoblotting with indicated antibodies. Error bars represent SD ($n = 3$). **$P < 0.01$ (Student's $t$-test).

I        JNK inhibition reduced ER stress-induced apoptosis of MITOL-KO MEFs. MEFs were treated with 20 μM JNK inhibitor II (JNK inh) 2 h prior to Tu treatment for 24 h,
         followed by immunoblotting with indicated antibodies. Error bars represent SD ($n = 3$). **$P < 0.01$ (Student's $t$-test).

its inactive mutant (Fig 4G). An *in vitro* ubiquitylation assay also confirmed the direct ubiquitylation of IRE1α by MITOL (Fig EV3A). These results demonstrate that IRE1α is a novel substrate for MITOL. Recent studies have identified various types of polyubiquitin chains and have shown that different linkage types of the polyubiquitin chain have different effects on substrates (Pickart & Fushman, 2004; Mukhopadhyay & Riezman, 2007). The K48-linked polyubiquitin chain primarily mediates the degradation of substrates via the proteasome, whereas the K63-linked polyubiquitin chain is involved in the regulation of the activity, localization, and binding partner of substrates. Interestingly, IRE1α ubiquitylation by MITOL was recognized with the K63-linked polyubiquitin chain-specific antibody (Fig 4H). Consistently, IRE1α ubiquitylation was observed when the wild-type ubiquitin or K48R ubiquitin mutant was co-expressed with MITOL; however, when the K63R ubiquitin mutant was expressed, MITOL failed to ubiquitylate IRE1α (Fig 4I). We previously constructed various ubiquitin mutants, including a K-all-R mutant lacking intact lysine and an RK mutant with only one lysine (Sugiura *et al*, 2013). IRE1α ubiquitylation by MITOL was confirmed in HEK293 cells transfected with the R63K ubiquitin mutant (Fig 4J). These results show that MITOL adds a K63-linked polyubiquitin chain to IRE1α. MITOL depletion did not affect the degradation of IRE1α following cycloheximide treatment (Fig EV3B); therefore, MITOL is considered to regulate IRE1α properties but not the stability of IRE1α.

We next examined whether IRE1α ubiquitylation changed during ER stress. Importantly, IRE1α ubiquitylation-mediated MITOL was rapidly reduced by prolonged tunicamycin treatment (Fig EV3C). Prolonged tunicamycin treatment also enhanced the interaction between BIM and IRE1α (Fig EV3D). This reduction in IRE1α ubiquitylation preceded an apoptotic response following tunicamycin treatment (Fig EV3E), suggesting that the loss of IRE1α ubiquitylation directs the apoptotic switch of IRE1α signaling. However, the activity of MITOL did not change in response to tunicamycin treatment (Fig EV3F). Upon ER stress, IRE1α initially undergoes self-association through its luminal domain, leading to trans-phosphorylation, and then, IRE1α interfaces via its cytosolic domain allowing for mRNA docking onto IRE1α and RNase activation. To understand the molecular mechanism behind the recognition of IRE1α by MITOL, cells were transfected with K121Y or D123P mutants of IRE1α, which lack the ability of luminal self-association (Zhou *et al*, 2006; Li *et al*, 2010). IRE1α ubiquitylation by MITOL was enhanced by the mutation of K121Y or D123P (Fig EV3G), suggesting that MITOL preferentially ubiquitylates

monomeric IRE1α. Similarly, treatment with 4μ8C, an inhibitor of IRE1α RNase, enhanced the MITOL-dependent IRE1α ubiquitylation and the interaction between MITOL and IRE1α (Fig EV3H and I). Since 4μ8C was reported to interact not only with K907 in the RNase domain but also with K599 in the kinase domain of IRE1α (Cross *et al*, 2012), it is possible that 4μ8C also inhibits the kinase activity of IRE1α. To confirm this, we used two IRE1α kinase inhibitors, APY-29 and KIRA6, which are competitive inhibitors at the ATP binding domain. Allosteric modulation caused by APY-29 binding has been previously found to lead to IRE1α oligomerization and increases in RNase activity, whereas the binding of KIRA6 with IRE1α inhibits both oligomerization and the RNase activity of IRE1α (Ghosh *et al*, 2014). Although KIRA6 increased IRE1α ubiquitylation by MITOL, APY-29 decreased IRE1α ubiquitylation (Fig EV3J). These observations suggest that MITOL specifically ubiquitylates the RNase-inactive form of IRE1α.

### Mutation of K481 in IRE1α enhances RIDD activity and leads to apoptosis

To identify the lysine residues required for IRE1α ubiquitylation by MITOL, we used an *in silico* search, UbPred. The analysis predicted that several lysine residues of IRE1α act as putative ubiquitin binding sites. Therefore, we generated three IRE1α mutants, in which the lysine residues predicted as IRE1α ubiquitylation sites were substituted with arginine. The IRE1α K481R mutant showed a significant reduction in ubiquitylation upon MITOL overexpression when compared to wild-type IRE1α and other KR mutants of IRE1α (Fig 5A). This mutation of K481R did not affect the interaction between MITOL and IRE1α (Fig 5B). These results indicate that MITOL adds a polyubiquitin chain specifically to K481 of IRE1α.

Spliced *xbp1* mRNA was normally observed in IRE1α-KO MEFs expressing either wild-type IRE1α or the K481R mutant (Fig EV4A), suggesting that the mutation of K481 in IRE1α does not cause dysfunction of IRE1α. Under basal conditions, IRE1α oligomerization and activation are inhibited, at least in part, by their association with a major ER chaperone BiP (Bertolotti *et al*, 2000). Upon accumulation of unfolded proteins, BiP dissociates from IRE1α and preferentially binds to unfolded proteins, allowing IRE1α activation. Therefore, IRE1α overexpression can activate only IRE1α signaling among the branched pathways of UPR without ER stress. Indeed, overexpression of either wild-type or the K481R mutant of IRE1α did not alter the autophosphorylation of PERK and ATF6 cleavage

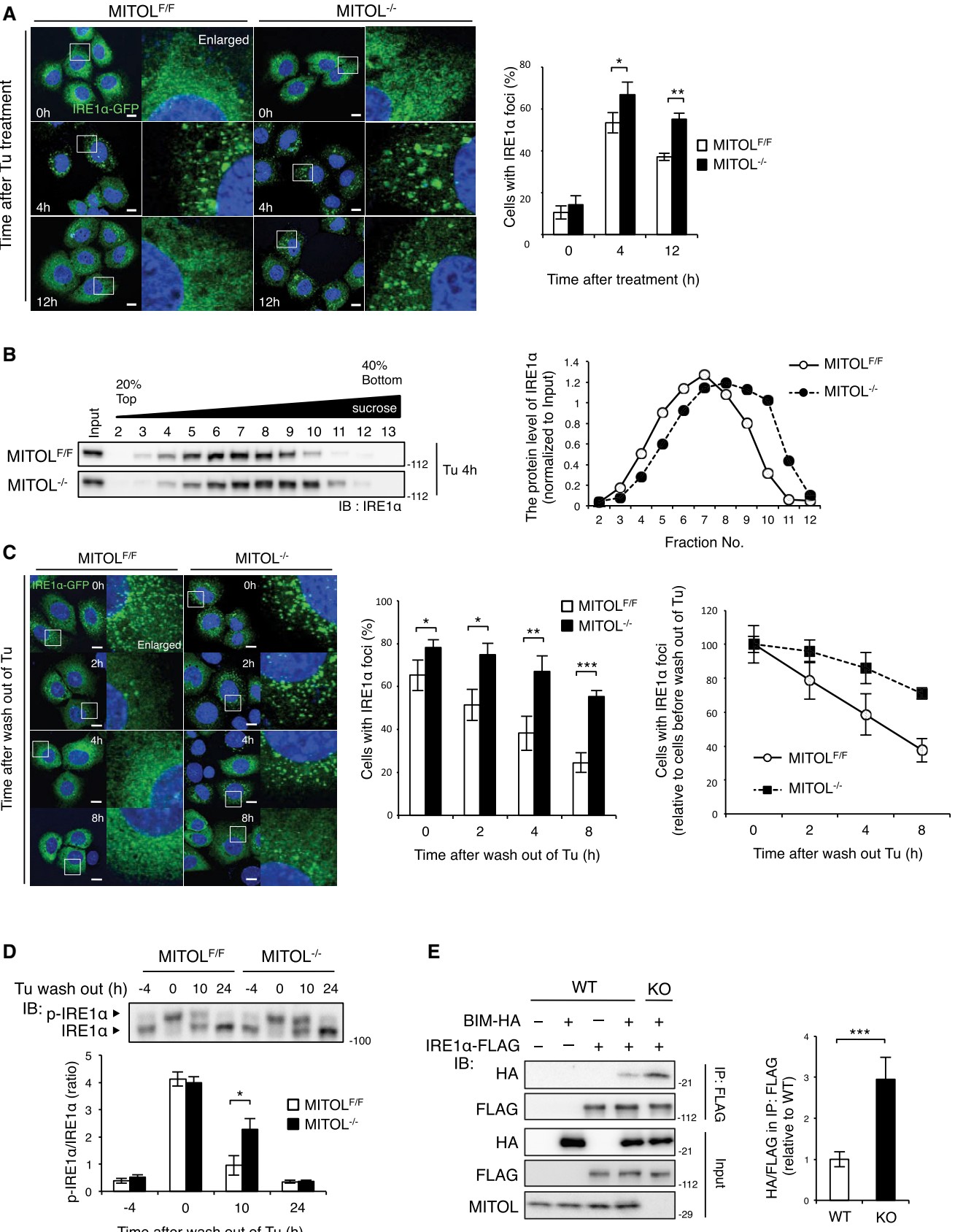

Figure 3.

**Figure 3. MITOL KO stabilizes IRE1α oligomers.**

A, B  MITOL KO facilitated IRE1α oligomerization. MEFs were transfected with IRE1α-GFP expression vector for 24 h prior to Tu treatment for indicated periods (A). The right panels show fivefold magnification images of the boxed regions. Percentages of cells with IRE1α foci were calculated from 100 cells by visual inspection in each independent experiment (A). To evaluate the oligomerization level of endogenous IRE1α, cells without any transfection were solubilized and separated by sucrose density-gradient centrifugation, followed by immunoblotting with anti-IRE1α antibody (B). Scale bar represents 10 μm. Error bars represent SD ($n = 4$). *$P < 0.05$, **$P < 0.01$ (Student's $t$-test).

C  IRE1α oligomers were stabilized by MITOL KO. MEFs transfected with IRE1α-GFP were treated with Tu. After 4 h, these cells were washed with PBS and subsequently re-fed with fresh media for indicated periods. The right panels show fivefold magnification images of the boxed regions. Percentages of cells with IRE1α foci were calculated from 100 cells by visual inspection at indicated periods after wash out of Tu in each independent experiment. Scale bar represents 10 μm. Error bars represent SD ($n = 3$). *$P < 0.05$, **$P < 0.01$, ***$P < 0.001$ (Student's $t$-test).

D  Sustained IRE1α autophosphorylation after wash out of Tu in MITOL-KO MEFs. MEFs were treated with Tu for 4 h and washed with PBS and re-fed with fresh media for indicated periods, followed by immunoblotting with anti-IRE1α antibody after Phos-tag SDS–PAGE. Error bars represent SD ($n = 3$). *$P < 0.05$ (Student's $t$-test).

E  MITOL deletion promoted the interaction between IRE1α and BIM under ER stress. MITOL-depleted HEK293 cells (KO) were generated using CRISPR-Cas9 system. Cells were co-transfected with indicated vectors 24 h prior to Tu treatment for 12 h. Cells were immunoprecipitated with anti-FLAG antibody, followed by immunoblotting with indicated antibodies. Error bars represent SD ($n = 3$). ***$P < 0.001$ (Student's $t$-test).

(Fig EV4B). We next examined whether the K481R mutant caused IRE1α hyper-activation and apoptosis. Overexpression of the IRE1α K481R mutant resulted in greater Annexin V-positive cells and cleaved caspase-3 levels than the overexpression of wild-type IRE1α or an empty vector (Fig 5C and D). Overexpression of the IRE1α K481R mutant decreased the mRNA levels of genes degraded by RIDD and increased *txnip* mRNA (Fig 5E and F). In addition, the IRE1α K481R mutant exhibited increased interaction with BIM and a higher oligomerization capability than that of wild-type IRE1α (Fig 5G and H). Moreover, the IRE1α K481R mutant showed a strong reduction of cell viability and a drastic induction of cell death upon tunicamycin treatment (Fig EV4C and D). These findings suggest that the K481 mutation in IRE1α leads to hyper-oligomerization of IRE1α and RIDD activation. Furthermore, the overexpression of IRE1α K481R led to significant morphological abnormalities in the ER and an increase of resting Ca$^{2+}$ levels in the ER compared to those of the empty vector and wild-type IRE1α (Fig EV4E and F). Thus, MITOL is a novel regulator for IRE1α.

## MITOL-mediated ubiquitylation of IRE1α is dependent on MAM formation

It was reported that both IRE1α and MITOL are partially localized at MAM (Mori *et al*, 2013; Sugiura *et al*, 2013). As previously reported, tagged MITOL showed a mitochondria-like structure, whereas tagged IRE1α was observed to form an ER-like structure (Fig EV5A). Since some regions of the ER network are connected to mitochondria, the tagged MITOL was partially co-localized with ER-like structure visualized by the tagged IRE1α (Fig EV5A). The merged image and line profile showed a partial co-localization of MITOL and IRE1α (Fig EV5A). A Percoll density-gradient experiment also indicated that endogenous MITOL and IRE1α are abundant in the MAM fraction (Fig 6A). A crude mitochondrial fraction collected by cell fractionation contains both mitochondrial and non-mitochondrial membranes such as MAM. Because MAM is a cholesterol-rich membrane, it is preferentially solubilized with a low dose of digitonin. Similar to CNX, both IRE1α and MITOL were solubilized from the crude mitochondrial fraction by treatment with 0.8 mg/ml digitonin; on the other hand, Tom20 was detected in precipitates (Fig 6B). These results show that IRE1α and MITOL are enriched at the MAM.

We next investigated whether MAM is required for IRE1α ubiquitylation by MITOL. MITOL and IRE1α were co-immunoprecipitated

in the crude mitochondrial fraction containing MAM (Fig 6C). MITOL overexpression enhanced IRE1α ubiquitylation only in the crude mitochondrial fraction containing MAM (Fig 6D). PACS2 is the first molecule identified to be involved in MAM formation, and PACS2 knockdown results in the dissociation between the ER and mitochondria (Simmen *et al*, 2005). The interaction between MITOL and IRE1α was inhibited by siPACS2 (Fig 6E). MITOL-mediated IRE1α ubiquitylation was also reduced by the suppression of PACS2 (Fig 6F). Similarly, knockdown of Mfn2, another protein involved in MAM formation, also blocked MITOL-mediated IRE1α ubiquitylation (Fig 6G). MITOL deletion or the K481R mutation of IRE1α did not alter the MAM localization of IRE1α (Fig EV5B and C). These results suggest that the association between IRE1α and MITOL is dependent on MAM.

## MITOL depletion results in IRE1α hyper-activation and leads to apoptosis in the spinal cord

To explore the physiological significance of IRE1α regulation by MITOL, we generated nerve-specific MITOL-KO mice, named MITOL[nestin], by crossing MITOL[F/F] mice with Nestin-Cre transgenic mice (Fig EV5D–F). The spinal cord of MITOL[nestin] mice displayed reduced IRE1α ubiquitylation (Fig 7A). We attempted to generate ER stress model mice via the intraperitoneal injection of tunicamycin. Sucrose density-gradient centrifugation analysis showed that the spinal cord of MITOL[nestin] mice exhibited IRE1α hyper-oligomerization 24 h after tunicamycin injection (Fig 7B). IRE1α-dependent *xbp1* splicing and mRNA expression of the downstream genes increased upon tunicamycin treatment in the spinal cord of MITOL[nestin] mice (Fig 7C). In contrast, the target genes of IRE1α-dependent mRNA decay were found to be downregulated (Fig 7D). Consistent with the results obtained from MITOL-KO MEFs, the mRNA levels of *txnip* were upregulated strongly in the spinal cord of MITOL[nestin] mice under ER stress (Fig 7E). On the other hand, MITOL depletion in the spinal cord did not affect PERK autophosphorylation and the induction of its downstream genes, *atf4* and *chop*, following tunicamycin treatment (Fig EV5G and H). Furthermore, the spinal cord of MITOL[nestin] mice showed a remarkably higher number of TUNEL-positive cells and lower numbers of Nissl-positive cells 48 h after tunicamycin injection compared to those of the wild-type mice (Fig 7F and G). These findings demonstrate that MITOL also inhibits the hyper-activation of IRE1α and subsequent cell death in the spinal cord under ER stress.

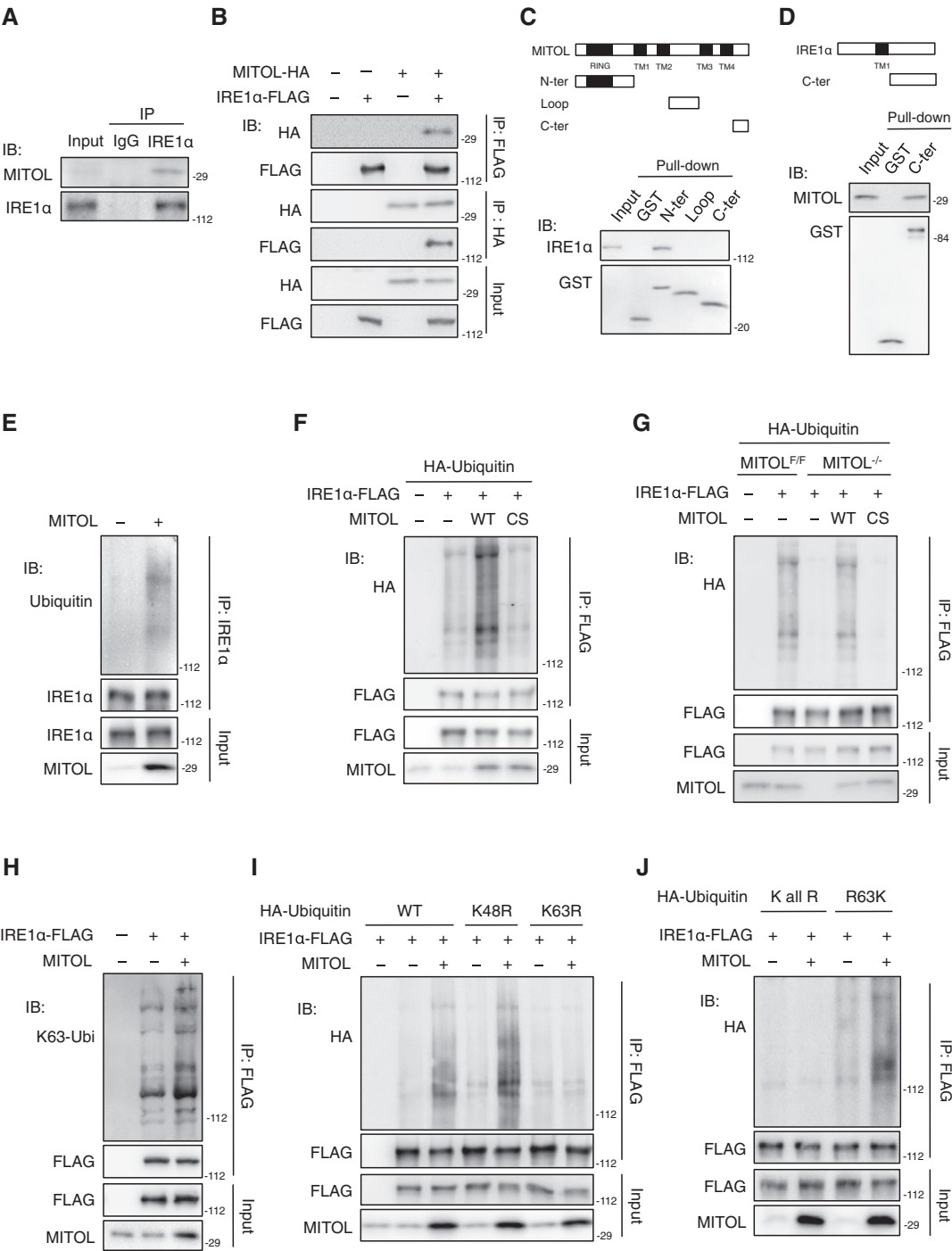

**Figure 4. MITOL interacts with and ubiquitylates IRE1α.**

A–D   Interaction between MITOL and IRE1α. Lysates of HEK293 cells were subjected to immunoprecipitation with anti-IRE1α antibody, followed by immunoblotting with indicated antibodies (A). Lysates of HEK293 cells co-transfected with indicated vectors were subjected to immunoprecipitation, followed by immunoblotting with indicated antibodies (B). GST pull-down assay of IRE1α or MITOL was performed on lysates of MEFs, followed by immunoblotting with indicated antibodies (C, D). N-ter: 1–90 aa of MITOL; Loop: 158–210 aa of MITOL; C-ter: 241–278 aa of MITOL, 465–977 aa of IRE1α.

E–G   MITOL ubiquitylated IRE1α. Lysates of HEK293 cells (E, F) or MEFs (G) transfected with indicated vectors were immunoprecipitated with indicated antibody, followed by immunoblotting with indicated antibodies.

H–J   MITOL adds lysine 63-linked polyubiquitin chains to IRE1α. Lysates of HEK293 cells transfected with indicated vectors were immunoprecipitated with indicated antibody. IRE1α ubiquitylation was detected as similar to (E). WT: wild-type ubiquitin. K48R, K63R: ubiquitin mutants containing lysine (K)-to-arginine (R) substitution at position 48 or 63, respectively. K all R: ubiquitin mutant containing all lysine-to-arginine substitutions. R63K: ubiquitin mutants containing only one lysine intact at position 63.

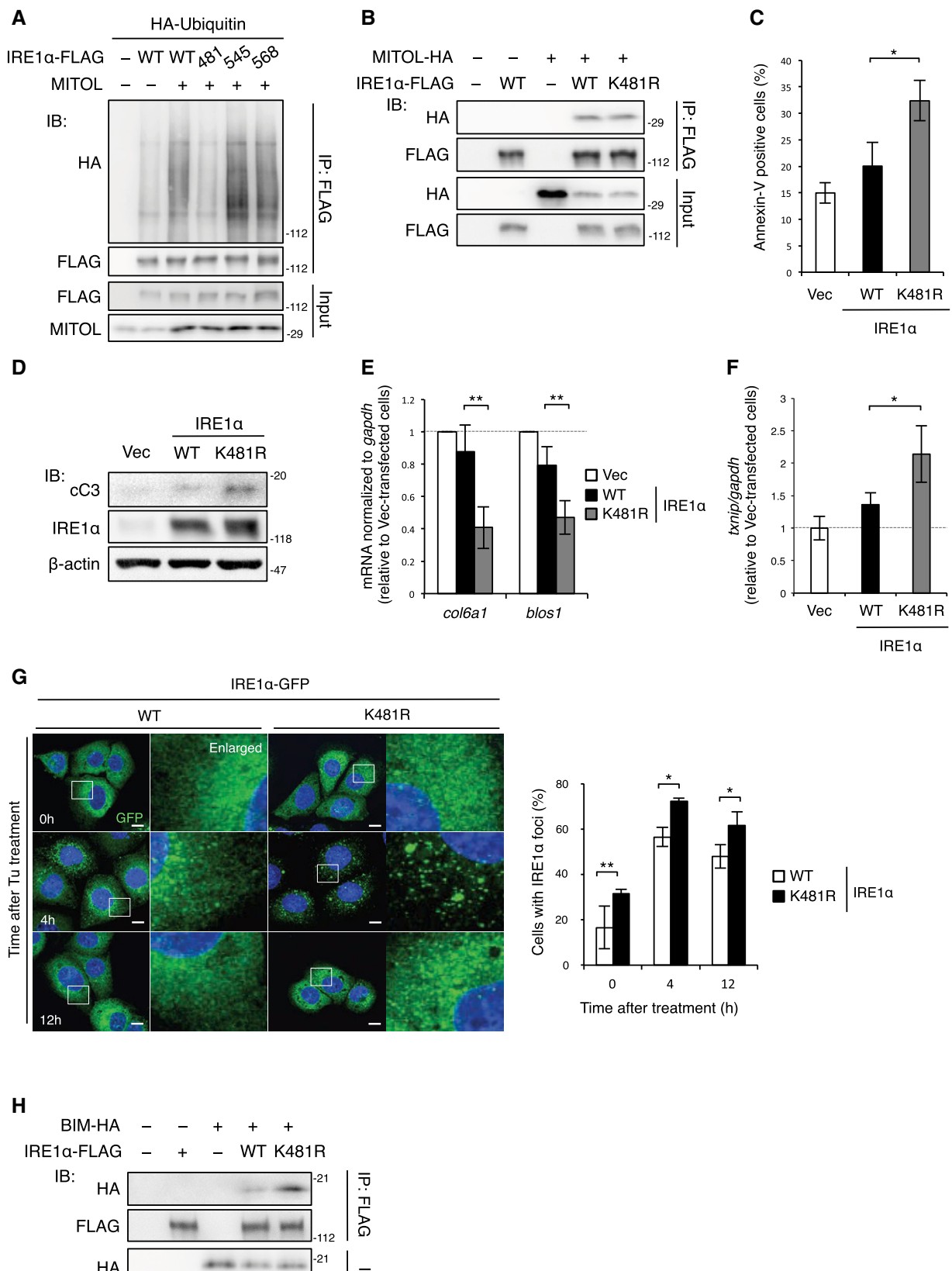

Figure 5.

**Figure 5.  IRE1α K481R induces apoptosis via abnormal clustering of IRE1α and RIDD.**

A       MITOL added polyubiquitin chains to K481 of IRE1α. Lysates of HEK293 cells transfected with each lysine mutant of IRE1α and indicated vectors were immunoprecipitated with anti-FLAG antibody, followed by immunoblotting with indicated antibodies. 481: K481R; 545: K545R; 568: K568R.

B       IRE1α K481R interacted with MITOL. Lysates of HEK293 cells transfected with indicated vectors were immunoprecipitated with anti-FLAG antibody, followed by immunoblotting with indicated antibodies.

C, D    Overexpression of IRE1α K481R-induced apoptosis. MEFs were transfected with indicated vectors. After 24 h, these cells were stained with Annexin V-FITC (C) or subjected by immunoblotting with indicated antibodies (D). Error bars represent SD ($n = 3$). *$P < 0.05$ (Student's $t$-test).

E, F    Enhanced RIDD activity in IRE1α K481R. MEFs were transfected with indicated vectors 24 h prior to analysis (E, F). The RIDD activity was measured by qRT–PCR. Error bars represent SD ($n = 3$). *$P < 0.05$, **$P < 0.01$ (Student's $t$-test).

G       Hyper-oligomerization of IRE1α K481R. MEFs were transfected with IRE1α-GFP or IRE1α K481R-GFP. After 24 h, these cells were exposed with Tu for indicated periods and GFP signals were observed. The right panels show 5.5-fold magnification images of the boxed regions. Percentages of cells with IRE1α foci were calculated from 100 cells by visual inspection in each independent experiment. Scale bar represents 10 μm. Error bars represent SD ($n = 3$). *$P < 0.05$, **$P < 0.01$ (Student's $t$-test).

H       Increased interaction of IRE1α K481R with BIM. Cells transfected with indicated vectors 24 h prior to immunoprecipitation, followed by immunoblotting with indicated antibodies.

## Discussion

IRE1α is a key sensor protein that determines cellular adaptation and cell death under ER stress. A recent study reported that the dimeric form of IRE1α is structurally suitable for *xbp1* splicing, which leads to cellular adaptation (Zhou *et al*, 2006). However, IRE1α hyper-oligomerization facilitates not only cellular adaption but also cell death via a variety of mRNA and miRNA decays, termed RIDD, and JNK phosphorylation (Ghosh *et al*, 2014). TXNIP and caspase-2 upregulation by RIDD are suggested to be involved in IRE1α-mediated apoptosis. IRE1α also phosphorylates ASK1 via the adaptor protein TRAF2, which induces JNK phosphorylation and apoptosis (Nishitoh *et al*, 2002). Here, we identified MITOL as a novel regulator of IRE1α (Fig 4). MITOL inhibited excessive oligomerization and activation of IRE1α by adding a K63-linked polyubiquitin chain to IRE1α (Figs 2–5). Although IRE1α is known to be ubiquitylated and degraded by HRD1 and TRAF6 (Qiu *et al*, 2013; Sun *et al*, 2015), our findings are the first to demonstrate the regulation of IRE1α activity by ubiquitylation. Several studies have shown that IRE1α oligomers are stabilized by the association with Bcl-2 family proteins, including BIM (Woehlbier & Hetz, 2011). Here, we also showed that MITOL depletion stabilized IRE1α oligomers and enhanced the interaction between IRE1α and BIM (Fig 3). Thus, it is possible that MITOL regulates the interaction between IRE1α and Bcl-2 family proteins to suppress the hyper-oligomerization of IRE1α and apoptosis induction under ER stress. A recent study has demonstrated that the basic residues in the juxtamembrane region of IRE1α cytosolic domain contribute to mRNA docking onto its oligomers. Since the basic residue K481 of IRE1α, a specific site ubiquitylated by MITOL, is located in the juxtamembrane region, MITOL may regulate IRE1α RNase activity via mRNA docking, in addition to the stability of its oligomers.

The ER and mitochondria are known to functionally cooperate through membrane contact, named MAM. Lipids and $Ca^{2+}$ are transferred between the ER and mitochondria (Raturi & Simmen, 2013). MAM is a membrane domain containing lipid rafts and is therefore suitable to function as a scaffold for signal regulation. MAM has been demonstrated to regulate intracellular signals, inducing inflammation and autophagy (Zhou *et al*, 2011; Hamasaki *et al*, 2013; Horner *et al*, 2015). However, the structural and functional changes of MAM under physiological and pathological conditions remain to be fully investigated. A recent study reported that MAM formation is upregulated under ER stress (Csordas *et al*, 2006). In addition, MAM-localized UPR sensors, PERK and IRE1α, induce an excessive influx of $Ca^{2+}$ from the ER to mitochondria, thereby promoting ER stress-dependent apoptosis (Verfaillie *et al*, 2012; Saida *et al*, 2014). Another study reported that the protein complex formation by CDIP1 and BAP31 is important for apoptotic signal transduction from the ER to mitochondria (Namba *et al*, 2013). These reports suggest that mitochondria induce $Ca^{2+}$-dependent apoptosis under terminal ER stress. Conversely, the role of mitochondria is poorly characterized in ER stress prior to initiating apoptosis. In this study, we showed that the modification of IRE1α by K63-linked ubiquitylation is MAM-dependent (Fig 6). Actually, the silencing of PACS2 or Mfn2, tethers of ER–mitochondria contacts, attenuates IRE1α ubiquitylation by MITOL. Therefore,

**Figure 6.  MAM is required for IRE1α ubiquitylation by MITOL.**

A       Accumulations of MITOL and IRE1α in the MAM fraction. Each organelle fraction was isolated from MEFs by Percoll density-gradient centrifugation, followed by immunoblotting with indicated antibodies. mito: pure mitochondria; micro: microsome; cyto: pure cytosol.

B       High solubility of MITOL and IRE1α in the low-dose digitonin. Raft-like structure including MAM was solubilized from the crude mitochondrial fraction of MEFs with indicated concentration of digitonin. S: supernatant; P: pellet.

C, D    IRE1α regulation by MITOL occurred in the MAM-rich mitochondrial fraction. Crude mitochondrial fraction containing MAM (Mito plus MAM) and the cytosolic fraction containing ER (Cyto plus ER) were isolated from HEK293 cells transfected with indicated vector, and subsequently, immunoprecipitation assay was performed to detect the interaction between MITOL and IRE1α (C) or the level of IRE1α ubiquitylation (D).

E       Knockdown of PACS2 impaired the interaction of MITOL with IRE1α. HEK293 cells were transfected with either scramble RNA or siPACS2 and indicated vectors for 24 h prior to immunoprecipitation assay.

F, G    Knockdown of PACS2 or Mfn2 inhibited IRE1α ubiquitylation by MITOL. HEK293 cells were transfected with either scramble RNA, siPACS2, or siMfn2 and indicated vectors 24 h prior to immunoprecipitation assay.

**A**

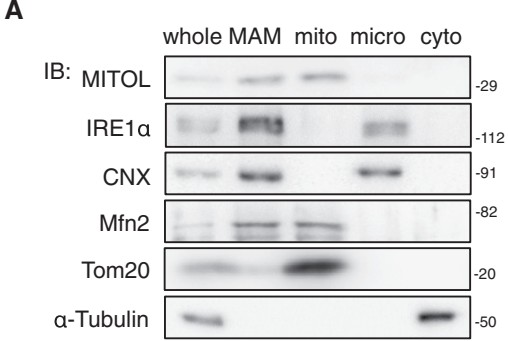

**B**

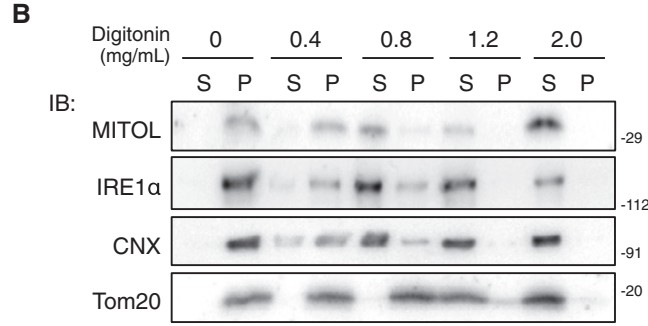

**C**

**D**

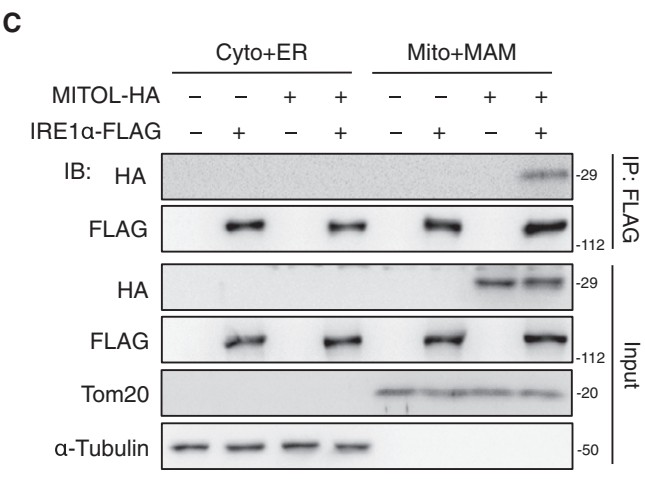

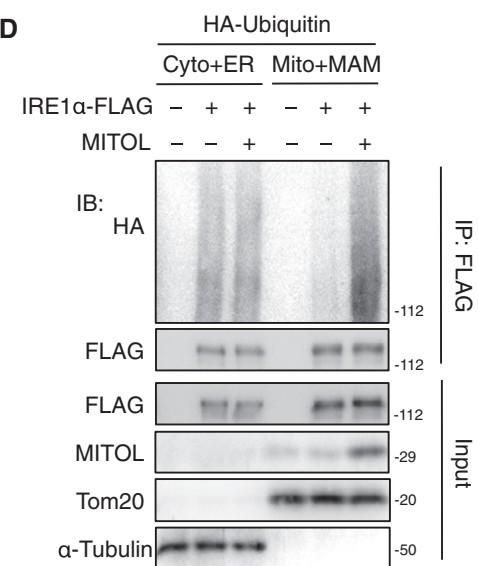

**E**          **F**          **G**

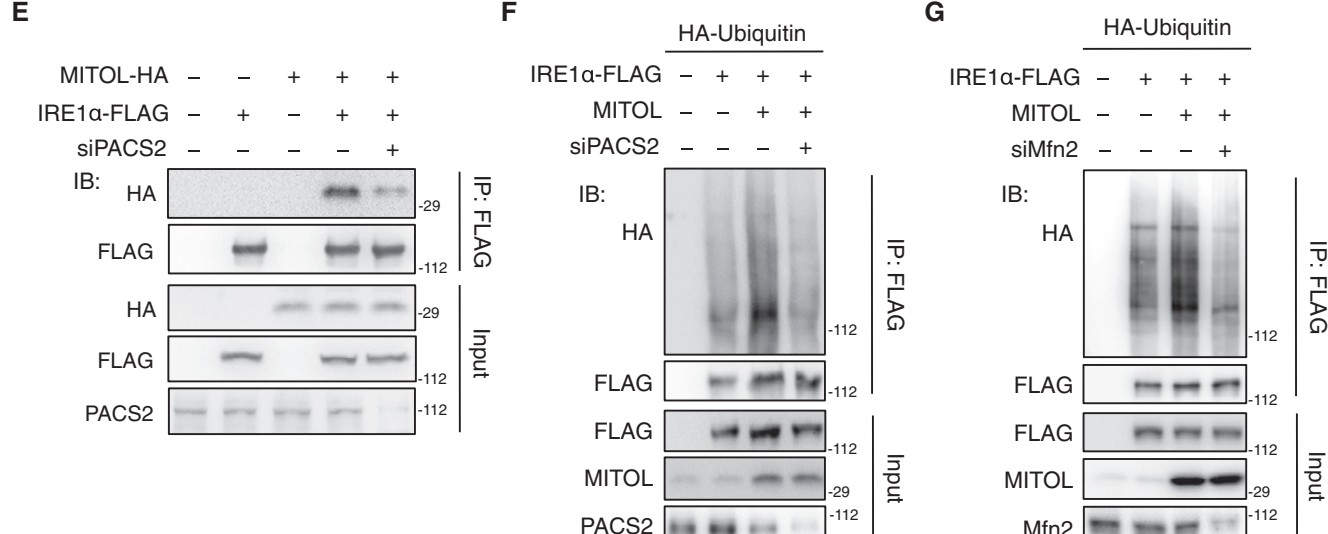

**Figure 6.**

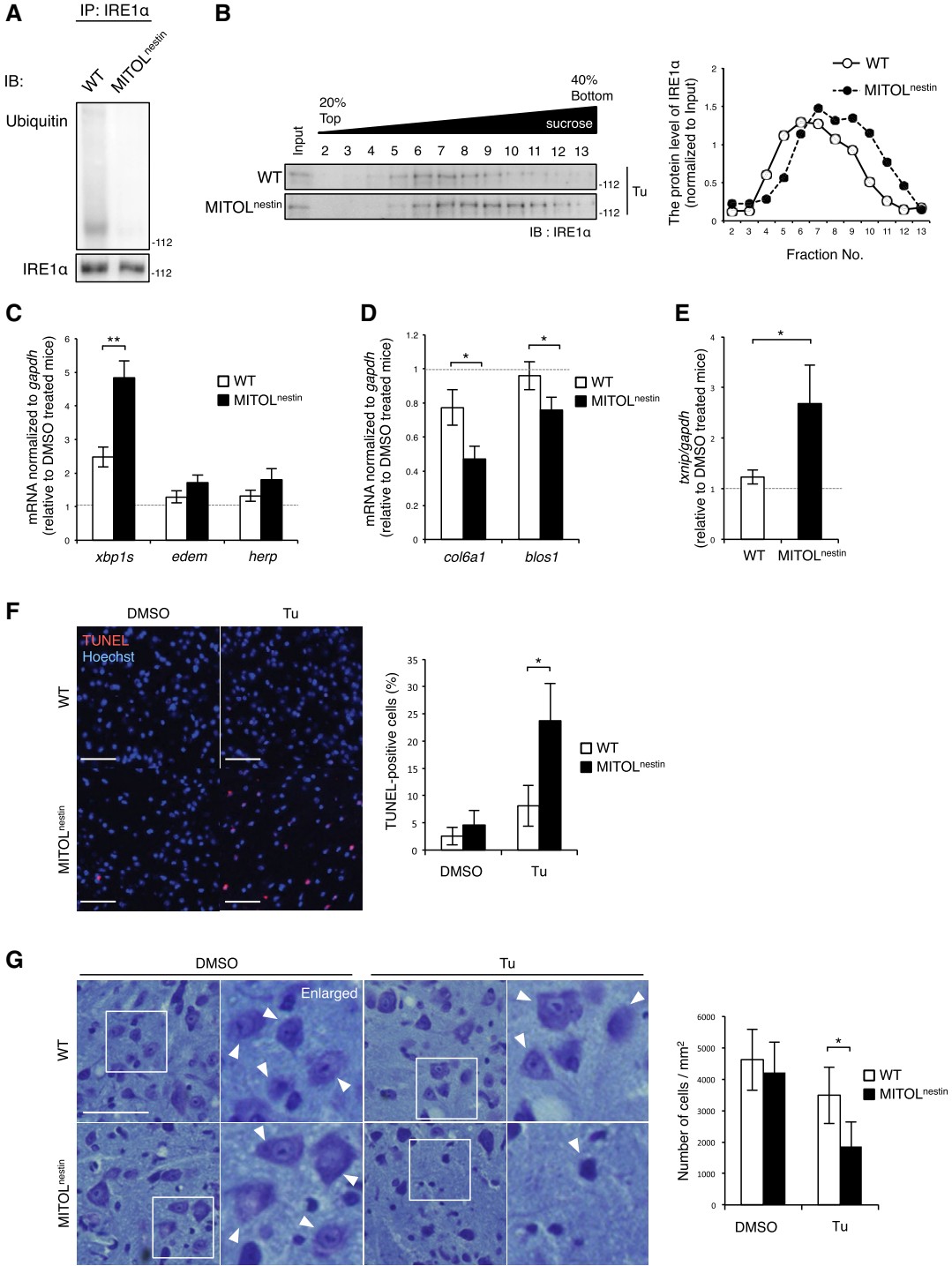

**Figure 7. IRE1α hyper-activation accompanying with apoptosis in the MITOL-KO spinal cord under ER stress.**

A   MITOL ubiquitylated IRE1α in the spinal cord of mice. Lysates of the spinal cord in mice were immunoprecipitated with anti-IRE1α antibody, followed by immunoblotting with indicated antibodies. WT: MITOL$^{F/F}$ mice; MITOL$^{nestin}$: MITOL$^{F/F, Nestin-Cre}$ mice.

B   Loss of MITOL led to hyper-oligomerization of IRE1α in the spinal cord under ER stress. Three-month-old WT or MITOL$^{nestin}$ mice were treated with 1 mg/kg Tu for 24 h. Spinal cord was solubilized with NP-40 lysis buffer and separated by sucrose density-gradient centrifugation.

C–E  MITOL deletion resulted in hyper-activation of IRE1α RNase in ER stress. WT or MITOL$^{nestin}$ mice were injected with Tu for 24 h, and spinal cord was analyzed by qRT–PCR. Error bars represent SD ($n = 3$). *$P < 0.05$, **$P < 0.01$ (Student's $t$-test).

F, G  Increased ER stress-induced apoptosis in the MITOL-KO spinal cord. Three-month-old WT or MITOL$^{nestin}$ mice were treated with Tu for 24 h, and each spinal cord was stained using TUNEL and Hoechst (F) or cresyl violet (G). The right panels show twofold magnification images of the boxed regions. Arrowheads indicate the representative neurons. Scale bar represents 50 μm (F) or 20 μm (G). Error bars represent SD ($n = 3$). *$P < 0.05$ (Student's $t$-test).

under ER stress, MAM plays not only an apoptotic role due to $Ca^{2+}$ overload in mitochondria but also a survival role through the inhibition of IRE1α hyper-activation. Notably, IRE1α ubiquitylation by MITOL was observed even under basal conditions and was markedly reduced following prolonged ER stress (Fig EV3). These findings suggest that MAM function dynamically changes in response to ER stress and that a rapid attenuation of MITOL-mediated IRE1α ubiquitylation under fatal ER stress may switch cell fate from cell survival to apoptosis. Although MAM formation under basal conditions or low levels of ER stress allows $Ca^{2+}$ influx into the mitochondria, this $Ca^{2+}$ influx is not sufficient to induce apoptosis. Additionally, the association between MITOL and IRE1α is maintained; as a result, IRE1α hyper-oligomerization and RIDD activity are inhibited. However, under chronic or high levels of ER stress, it is hypothesized that the disruption of MITOL-mediated IRE1α inhibition leads to rapid apoptosis, which is caused by IRE1α-dependent excessive $Ca^{2+}$ influx into mitochondria and is mediated by the bifunctional activity of IRE1α. So far, the mechanistic understanding of converting the two outputs of the UPR, cellular adaptation and cell death, has not been well characterized. Although DR5, a receptor for extrinsic apoptosis, was reported to play a central role in switching between cellular adaptation and cell death (Lu *et al*, 2014), a marked increase or decrease in DR5 levels was not observed in MITOL-KO MEFs compared to control MEFs (Fig EV2D). Therefore, MAM may be a new regulatory mechanism that switches between the two outputs of the UPR independently of DR5. However, to verify our hypothesis, further studies focusing on MAM are needed.

Induction of cell death under chronic ER stress contributes to the development of various diseases, such as cancer, heart failure, diabetes mellitus, and neurodegenerative diseases (Yoshida, 2007). Given that a fragmented or dilated ER is observed in amyotrophic lateral sclerosis (ALS) patients and in ALS model mice (Dal Canto & Gurney, 1995; Oyanagi *et al*, 2008; Lautenschlaeger *et al*, 2012), ER stress is suggested to be associated with ALS. Interestingly, IRE1α protein levels are increased prior to the development of ALS (Atkin *et al*, 2006, 2008). Therefore, IRE1α hyper-activation is considered likely to induce apoptosis in ALS patients. Here, we demonstrated that the MITOL-modulated IRE1α repression plays a survival role in spinal cord cells (Fig 7). Importantly, the association between MITOL and IRE1α is attenuated by prolonged ER stress (Fig EV3). Therefore, in ALS patients, the IRE1α-MITOL axis is likely to be negatively regulated, which leads to IRE1α hyper-activation and neuronal cell death. More detailed studies are required to clarify the mechanism for downregulating the IRE1α-MITOL axis under unresolved ER stress conditions. The inhibitors of this downregulation system or the activators of the IRE1α-MITOL axis may provide a new therapeutic strategy against ER stress-related diseases, including ALS.

# Materials and Methods

## Antibodies and regents

Anti-MITOL rabbit polyclonal antibodies were produced as described previously (Yonashiro *et al*, 2006). Anti-α-tubulin, anti-β-actin, and anti-FLAG-M2 antibodies were purchased from Sigma. Anti-cleaved caspase-3 (cC3), anti-cleaved PARP (cPARP), anti-

IRE1α, anti-PHB1, anti-SAPK/JNK, anti-pSAPK/JNK (T183/Y185), anti-PERK, and anti-pPERK (Thr980) antibodies were from Cell Signaling Technology. Anti-calnexin, anti-cytochrome c, anti-DRP1, and anti-PDI antibodies were from BD Biosciences. The anti-calnexin antibody purchased from BD Biosciences was used in Figs 6A and EV5B and C. Anti-Mfn2, anti-Tom20, anti-calnexin, anti-ATF6, and anti-BAX antibodies were from Santa Cruz Biotechnology. Anti-ubiquitin and anti-GFP antibodies were from MBL. Anti-HA was from COVANCE. Anti-PACS2 and anti-Tom20 antibodies were from Proteintech. The anti-Tom20 purchased from Proteintech was used in Fig EV1. Tunicamycin was obtained from Wako. Brefeldin A (Br) was from Focus Biomolecules. Tunicamycin (Tu) was dissolved in DMSO and used for cellular experiments at 0.7 μg/ml. DMSO was also treated with cells as control. Thapsigargin (Tg) was from Nacalai. Cycloheximide (CHX) and 4-hydroxytamoxifen (4-OHT) were from Sigma. 4-OHT was dissolved in ethanol (EtOH) and used for cellular experiments at 0.4 μM. Ethanol was also treated with cells as control. 4μ8C was from Millipore. APY-29 was from ChemScene. KIRA6 was from Cayman Chemical. JNK inhibitor II (JNK inh) was from Calbiochem. Ionomycin was from Wako.

## DNA constructs

MITOL expression vectors and ubiquitin mutants were constructed as described previously (Sugiura *et al*, 2013). IRE1α-FLAG and IRE1α-HA expression vector were obtained by subcloning IRE1 alpha-pcDNA3.EGFP (purchased from Addgene). Point mutations of IRE1α were generated with the site-directed mutagenesis kit (Stratagene). IRE1α-GFP expression vector was generated as described previously (Li *et al*, 2010). miRNA luciferase reporters, G-CEPIA1er, and mCherry-Sec61β (mC-Sec61β) were purchased from Addgene.

## Cell culture and transfection

MEFs, HEK293, and COS-7 cells and HeLa cells were grown in Dulbecco's modified Eagle's medium (DMEM) supplemented with 10% fetal bovine serum (FBS) and penicillin/streptomycin. Cells were transfected with either Lipofectamine 3000 (Invitrogen) or RNAiMax (Invitrogen) according to the manufacturer's protocol. The following siRNAs were used: siIRE1α #1: sense, 5′-CGGGCUC-CAUCAAGUGGACUUUAAATT-3′, antisense, 5′-UUUAAAGUCCA-CUUGAUGGAGCCCGTT-3′; siIRE1α #2: sense, 5′-AAGAUGGACUGG CGGGAGATT-3′, antisense, 5′-UCUCCCGCCAGUCCAUCUUTT-3′; siJNK1 #1: sense, 5′-AAAGAAUGUCCUACCUUCUUCUT-3′, antisense, 5′-AGAAGGUAGGACAUUCUUUTT-3′; and siJNK1 #2: sense, 5′-GCAGAAGCAAACGUGACAACATT-3′, antisense, 5′-UGUUGUCAC-GUUUGCUUCUGCTT-3′. siPACS2 and siMfn2 were described previously (Arasaki *et al*, 2015). The CRISPR Design Tool (http://www.genome-engineering.org/crispr/) was used to select the genomic sequence target in human MITOL (5′-CCAAGCCCTACAGCAGATGC-3′). Oligo pairs encoding 20-nt guide sequences were annealed and ligated into the plasmid pX330 (purchased from Addgene). HEK293 cells were transfected and MITOL-KO clones selected by serial dilution.

## Subcellular fractionation

Isolation of crude mitochondria or pure MAM was performed as described previously (Sugiura *et al*, 2013). Briefly, cells were

homogenized with homogenization buffer (HB) (5 mM HEPES pH 7.4, 0.5 mM EDTA, and 250 mM mannitol) and the crude mitochondrial fraction (pellet after centrifugation at 8,000 × *g*) was subjected to further separation by 30% Percoll gradient centrifugation at 95,000 × *g* for 30 min at 4°C. A low-density band (denoted as the MAM fraction) was purified. The pure mitochondrial fraction in the high-density band was resuspended in HB and pelleted by centrifugation (6,300 × *g* for 10 min). Microsomes were pelleted from the mitochondrial supernatant by centrifugation at 100,000 × *g* for 1 h. Cytosol was recovered as the supernatant from this centrifugation.

## RNA isolation and qRT–PCR

Total RNA was isolated from mammalian cells using RNeasy kit (Qiagen) and subjected to reverse transcription to cDNA using ReverTra Ace qPCR RT Kit (TOYOBO), following the manufacturer's protocol. PCR was performed using a THUNDERBIRD SYBR qPCR Mix (TOYOBO). The PCR conditions were as follows: 95°C for 1 min followed by 40 cycles at 95°C for 15 s, 60°C for 30 s, and 72°C for 60 s. RT–PCR was performed using miScript SYBR Green PCR Kit and miScript Primer Assays (Qiagen).

The following primers were used: *atf4:* forward, 5′-GGA CAGAT TGGATGTTGGAGAAAATG-3′, reverse, 5′-GGAGATGGCCAATTGGG TTCAC-3′; *caspase-2:* forward, 5′- CCACAGATGCTACGGAACA-3′, reverse, 5′- GCTGGTAGTGTGCCTGGTAA-3′; *chop:* forward, 5′-CATACACCACCACACCTGAAAG-3′, reverse, 5′-CCGTTTCCTAGTT CTTCCTTGC-3′; *dr5:* forward, 5′-CTGTGCTACAGGCTGTCTTTG-3′, reverse, 5′-GTACTGGCCTGCTAGACAG-3′; *xbp1s:* forward, 5′-GTGTCAGAGTCCATGGGA-3′, reverse, 5′-GAGTCCGCAGCAGG-TG-3′; *edem:* forward, 5′-AAGCCCTCTGGAACTTGCG-3′, reverse, 5′-AACCCAATGGCCTGTCTGG-3′; Sec61: forward, 5′-CTATTTC CAGGGCTTCCGAGT-3′, reverse, 5′-AGGTGTTGTACTGGCCTCGGT-3′; *herp:* forward, 5′-CATGTACCTGCACCACGTCG-3′, reverse, 5′-GAGGACCACCATCATCCGG-3′; *col6a1:* forward, 5′-TGCTCAA CATGAAGCAGACC-3′, reverse, 5′-TTGAGGGAGAAAGCTCTGGA-3′; *blos1:* forward, 5′-CAAGGAGCTGCAGGAGAAGA-3′, reverse, 5′-GCCTGGTTGAAGTTCTCCAC-3′; *hgsnat:* forward, 5′-TCTCCG CTTTCTCCATTTTG-3′, reverse, 5′-CGCATACACGTGGAAAGTCA-3′; *pdgfrp:* forward, 5′-AACCCCCTTACAGCTGTCCT-3′, reverse, 5′-TAATCCCGTCAGCATCTTCC-3′; *txnip:* forward, 5′-TCAAGGGC CCCTGGGAACATC-3′, reverse, 5′-GACACTGGTGCCATTAAGTCAG-3′; and *gapdh:* forward, 5′-AACTTTGGCATTGTGGAAGG-3′, reverse, 5′-GGATGCAGGGATGATGTTCT-3′. Primers for *xbp1s* were designed to amplify the region containing sequences removed by IRE1α.

## Digitonin treatment

To extract cholesterol-enriched lipid raft from MAM, 100 μg of crude mitochondria was incubated for 10 min with indicated final concentrations of digitonin on ice. After centrifugation at 8,000 × *g*, the precipitate and supernatant were analyzed by immunoblotting.

## Immunoprecipitation

To examine the protein interaction, cells were solubilized with NP-40 lysis buffer (1% NP-40, 10 mM Tris–HCl pH 7.4, 150 mM NaCl, 0.5 mM EDTA, 10 mM NaF, and protease inhibitors) and

centrifuged at 20,000 *g* for 20 min at 4°C. The supernatant was subjected to immunoprecipitation using indicated antibodies. To evaluate the ubiquitylation level of proteins, cells were solubilized with RIPA lysis buffer (0.1% SDS, 0.05% DOC, 1% Triton X-100, 10 mM Tris–HCl pH 7.4, 150 mM NaCl, 5 mM EDTA, and protease inhibitors) and centrifuged at 20,000 *g* for 20 min at 4°C. The supernatant was sonicated 10 s and subjected to immunoprecipitation using indicated antibodies.

## Immunoblotting and Phos-tag PAGE

Whole lysates were separated by SDS–PAGE and transferred to the PVDF membranes (Millipore). The blots were probed with indicated antibodies, and protein bands on the blot were visualized by the enhanced chemiluminescence reagent (Millipore). Phos-tag-PAGE was performed using 7% SDS–PAGE minigels containing 10 μM Phos-tag (Wako) in the presence of 100 μM $MnCl_2$ following the manufacturer's protocol.

## *In vitro* ubiquitylation assay

IRE1α-FLAG and MITOL-HA were purified from HEK293 cells by immunoprecipitation. The purified proteins were incubated with reaction buffer containing 50 mM Tris–HCl, pH 7.4, 2 mM $MgCl_2$, 4 mM ATP, 200 ng of E1 (Nacalai), 600 ng of UbcH5b (Biomol), and 5 μg of His-Ub (Biomol) for 2 h at 30°C and then terminated with 3 × SDS sample buffer.

## GST pull-down assay

To determine the specific domains required for the interaction between MITOL and IRE1α, GST-fusion deletion mutants were prepared using glutathione high-capacity magnetic agarose beads (Sigma) as described previously (Yonashiro *et al*, 2006). Lysates from MEFs were incubated with the purified GST-fusion proteins. Precipitates were analyzed by immunoblotting.

## Immunofluorescence

Cells were fixed with 4% paraformaldehyde in PBS for 20 min at room temperature, washed twice with PBS and permeabilized with 0.1% Triton X-100 in PBS for 15 min, and then washed two times with PBS and blocked with 1% bovine serum albumin in PBS, all at room temperature. For double staining, the cells were incubated with indicated primary antibodies for 1 h at room temperature, washed 3 times with PBS, and then incubated with appropriate secondary antibodies for 30 min. The samples were washed as described above, mounted using Fluorescent Mounting Medium (Dako), and analyzed using an Olympus IX81 confocal fluorescence microscope. For classification of the morphology of mitochondrial network, four stacks of 0.2 μm were acquired for cells stained with anti-Tom20 antibody using 60× objective. Images were compiled and classified in a blinded manner as normal in which the majority of mitochondria were connected in the cell or abnormal in which many mitochondria exhibited the spherical structure or showed the disconnected network overall. For quantification of the morphology of the ER network, four stacks of 0.2 μm were acquired for cells transfected with mC-Sec61β using

60× objective and classified in a blinded manner as normal morphology with reticular-like ER network or abnormal morphology either with a more sheet-like ER network overall or with at least three aggregated parts in the ER network. Percentages of cells with abnormal ER network in each condition were calculated from 50 cells in each independent experiment.

## Immunohistochemistry

Spinal cord was fixed with 10% formalin neutral buffer solution (Wako), and 5-μm-thick paraffin sections were prepared. TUNEL staining was performed using *In Situ* Cell Death Detection Kit (Roche). For Nissl, staining was performed using 0.5% cresyl violet. The sections were observed using an All-in-One Fluorescence Microscope BZ-9000.

## Ca$^{2+}$ imaging

ER Ca$^{2+}$ imaging was performed as described previously (Suzuki *et al*, 2014; Hirabayashi *et al*, 2017). Briefly, MEFs were replaced on glass bottom dishes and transfected with G-CEPIA1er 24 h before Ca$^{2+}$ imaging. First, the medium was washed using HBSS (Gibco) and re-fed with HBSS. G-CEPIA1er signals of transfected cells were monitored per 500-ms interval during 60 s. At 20 s after Ca$^{2+}$ imaging started, these cells were co-incubated with 5 μM ionomycin. The level of Ca$^{2+}$ in the ER was obtained as $\Delta F$ values, calculated from $F_0$ minus $F_{max}$. $F_0$ values were obtained by the average intensity of G-CEPIA1er in each cell before ionomycin treatment. $F_{max}$ was calculated from the average intensity of G-CEPIA1er in each cell treated with ionomycin. The intensities of G-CEPIA1er were measured using ImageJ.

## Sucrose density-gradient assay

Sucrose gradient assay was performed as described previously (Area-Gomez *et al*, 2012). In brief, cells treated with tunicamycin for 4 h or mice treated with tunicamycin for 1 day were lysed for 20 min with NP-40 lysis buffer on ice and centrifuged at 20,000 *g* for 20 min at 4°C. The supernatant was centrifuged at 250,000 *g* for 12 h at 4°C through 20–40% sucrose gradient (150 mM NaCl, 1 mM EDTA, 50 mM Tris–HCl pH 7.4, and protease inhibitors), prepared freshly by progressively layering higher to lower density sucrose fractions in 5% increments. Each 4 ml gradient was divided evenly into 16 fractions (250 μl each), and then, aliquots of fractions 2–13 were analyzed by immunoblotting.

## Mice

MITOL$^{Flox/FLox}$ mice were crossed with Nestin-Cre transgenic mice. Genotypes were confirmed by tail tipping mice at around 21 days. Mice were genotyped using PCR primers #1, 5′-CACAGGTACGG TAGGTGTGTAAGC-3′, and primer #2, 5′-ATGGGAATGTGGTT CAGTTGTACC-3′. All animals were maintained under university guidelines for the care and use of animals. The experiments were performed after securing Tokyo University of Pharmacy and Life Sciences Animal Use Committee Protocol approval. For ER stress induction in the spinal cord, 3-month-old mice received intraperitoneal injection with either DMSO or tunicamycin (1 mg/kg) in saline.

## Flow cytometry

Annexin V-FITC staining was performed with Annexin V-FITC apoptosis detection kit (BioVision) according to the manufacturer's protocol. For the detection of mitochondrial depolarization, cells were incubated with 10 nM tetramethylrhodamine methyl ester (TMRM) for 30 min at 37°C and then washed with PBS. The fluorescence of TMRM was analyzed by flow cytometer (Becton Dickinson).

## Luciferase reporter assay

Luciferase assay was performed with dual-luciferase reporter assay system (Promega) according to the manufacturer's protocol.

## Cell viability assay and cell toxicity assay

Cell viability and cell toxicity were monitored using the Cell Counting Kit-8 (Dojindo) and Cytotoxicity LDH Assay Kit-WST (Dojindo), respectively, according to the manufacturer's protocol.

## Statistical analyses

All results are expressed as mean ± SD. Obtained data were compared between independent experiments using either two-tailed Student's *t*-test. The number of independent experiments is shown as *n*.

**Expanded View** for this article is available online.

## Acknowledgements

We thank Yuuji Otokuni, Kohei Arasaki, and Ayumu Sugiura for technical assistance. This study was supported in part by MEXT/JSPS KAKENHI (to S.N, R.I, T.F, and S.Y) and MEXT-Supported Program for the Strategic Research Foundation at Private Universities (to S.N, R.I, and S.Y), The Uehara Memorial Foundation, The Naito Foundation (to S.N and S.Y), The Takeda Science Foundation, The Sumitomo Foundation, The Cosmetology Research Foundation, The Ono Medical Research Foundation (to S.Y), The Tokyo Biochemical Research Foundation, and AMED under Grant Number JP17gm5010002.

## Author contributions

KT and SY designed and carried out the experiments. SN analyzed the data. AU carried out qRT–PCR and cultured cell analysis with contribution from IS and NI. SI generated MITOL$^{Flox/Flox}$ mice and MITOL-related MEFs. TI and TU generated IRE1α-KO MEFs. TT, TF, and NM contributed to the understanding of MITOL$^{nestin}$ mice phenotype. KT and SY wrote the manuscript with contribution from RI.

## Conflict of interest

The authors declare that they have no conflict of interest.

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
