## [Review Process File · The EMBO Journal]

MITOL prevents ER stress-induced apoptosis by IRE1 α ubiquitylation at ER-mitochondria contact sites

Keisuke Takeda, Shun Nagashima, Isshin Shiiba, Aoi Uda, Takeshi Tokuyama, Naoki Ito, Toshifumi Fukuda, Nobuko Matsushita, Satoshi Ishido, Takao Iwawaki, Takashi Uehara, Ryoko Inatome and Shigeru Yanagi.

Review timeline:

Submission date:	24 th October 2018
Editorial Decision:	6 th December 2018
Revision received:	15 th March 2019
Editorial Decision:	10 th April 2019
Revision received:	1 st May 2019
Accepted:	15 th May 2019

Editor: Elisabetta Argenzio

Transaction Report:

1st Editorial Decision

6th December 2018

Thank you for submitting your manuscript on a role for MITOL in preventing ER stress-induced apoptosis to The EMBO Journal. Your study has been sent to three referees for evaluation, and we have now received reports from them, which are enclosed below for your information.

As you can see, the referees concur with us on the potential interest of your findings. However, they also raise several critical points that need to be solved before they can support publication in The EMBO Journal. Importantly, the referees are concerned that the mechanistic aspects of MITOL-mediated protection on ER stress-induced apoptosis and IRE1 α oligomerization would need to be better characterized and request additional experiments and controls to strengthen the key conclusions.

Addressing these issues through decisive additional data as suggested by the referees would be essential to warrant publication in The EMBO Journal. Given the overall interest of your study, I would like to invite you to submit a revised version of the manuscript according to the referee's requests. I should add that it is The EMBO Journal policy to allow only a single round of revision, and acceptance of your manuscript will therefore depend on the completeness of your responses in this revised version.

REFeree REPORTS

Referee #1:

The manuscript by Takeda and co-authors addresses the role of the mitochondrial ubiquitin ligase (MITOL) in modulating the cellular response to ER stress through the regulation of IRE1 α . Previous work from the same group has shown that the mitochondrial ligase MITOL (March5) mediates K63-linked Ub of mitochondria associated Mfn2 at the MAMs thereby regulating ER-Mito tethering. They now show that IRE1 α is a novel substrate of MITOL. K63-linked polyubiquitination of IRE1 α

at K481 by MITOL prevents IRE1a hyper-oligomerization and RIDD activity, thereby reducing cell death (presumably apoptosis) in response to tunicamycin. Interestingly, absence of MITOL both in vitro as well as in vivo (in MITOL-deficient mouse spinal cord) tunicamycin (Tu) elicits enhanced RIDD activity and cell death. Finally the authors claim that the role of MITOL in modulating IRE1-mediated cell death pathways during ER stress is caused by MITOL-mediated ER-mitochondria interaction, thereby assessing the role of MAMs in protecting ER stress-dependent cell death through inhibiting IRE1a hyper-oligomerization. The manuscript is well structured and contains a lot of data which support in large part, the conclusions drawn by the authors. Yet, presumably into the attempt to fit a lot of data into a common paradigm, some mechanistic aspects of this interesting work remain rather preliminary. Moreover, in some experimental settings important controls (both positive and negative) are missing.

General comments:

Authors claim that 4-OHT induced MITOL loss induces 'abnormalities' in the mitochondria and ER network. However, from the staining shown in Figure S1A and S1B it is not clear how authors quantify these 'abnormalities'. Since persistence of a 'disturbed' ER morphology (fragmentation? not clear, how do the authors quantify these changes?) is observed either acute or chronic loss of MITOL, this requires a better assessment both at the morphological (e.g. using better markers of tubular and luminal ER like e.g. Sec61b and KDEL) and functional levels. This is important also considering that affecting ER-mito contact sites through MITOL expression may have implication in fundamental ER biological functions (e.g. is the ER-Ca²⁺ steady state level affected?). Moreover, can the authors explain why shDrp1 does not rescue completely the mitochondrial abnormalities?

The mechanistic aspects on MITOL-mediated protection on ER stress induced apoptosis would need some improvements and further controls. In general, authors should prove their point that MITOL depletion increases specifically ER stress induced apoptotic, therefore caspase-inhibitable cell death, by showing that MITOL then does not affect non-ER stress triggered apoptosis (e.g. by staurosporine or death receptor ligands).

Secondly under severe conditions of ER stress induction mitochondrial apoptosis (by engaging BH3 only proteins like BIM and NOXA and Bax/Bak mediated pore forming activity rather than inducing MPT), is regulated by IRE1a RIDD activity towards miRs-17 that represses translation of caspase-2, leading to Bid-mediated and Bax/Bak-induced cytochrome C release. Since MITOL deletion causes a reduction of miR-17 upon Tu treatment, the authors could test the involvement of caspase-2 in the mechanism of MITOL-regulated IRE1a dependent cell death.

Moreover, it would be interesting to analyse what happens to the interaction between BIM and IRE1a, and to the status of IRE1a-Ub under conditions of ER stress. Since the oligomerization of IRE1 is induced by ER stress, and lack of MITOL induces a hyper-oligomerization of IRE1a, is the interaction between BIM and IRE1 (especially in MITOL KO) increased after Tu, when IRE1 signaling is pro-apoptotic?

Related to Fig. 2, what is the effect of MITOL loss on UPR signalling at later time points? This point seems relevant since at the early phase of ER stress p-JNK inhibits cell death instead of promoting it (see paper of Brown et al, J Cell Science 2016). Also JNK inhibitors are known to have different off-target effects. Thus the role of JNK in MITOL-mediated IRE1 signalling should be better validated by a genetic approach.

Although both MITOL and IRE1a have been shown in other studies to be enriched at the MAMs, the hypothesis that MAMs integrity is required for the MITOL-IRE1 interaction and MITOL-mediated Ub, require a more rigorous analysis. For example Fig 6A; the staining of two proteins located at the ER and mitochondria, even if not necessarily enriched at the MAMs, will reveal a partial co-localization because the two organelles are interconnected. Thus this assay does not really prove the presence of these proteins at the MAMs. In Fig6B: CLX is not enriched in the MAM fraction and the mitochondria are contaminated by Tubulin and IRE1. IRE1 is also not so much present in the ER fraction. Also in Fig6E: The over-expression of MITOL-HA is not the same in the input, and it is actually much less in the PACS siRNA condition, thereby it is difficult to compare the two co-IP and to draw conclusions. Importantly, does the IRE1a K81R mutant show an impaired localization at the MAMs? And do conditions of ER stress weaken the MITOL-IRE1 interaction at

the ER-mitochondria contacts?

Specific remarks:

-Is the analysis presented in Fig.1C from the same WB? In the WB shown it seems that the MITOL and B-actin bands differ quite strongly, in comparison to the upper bands (cPARP and cC3); not clear also why the molecular weight of the CS mutant, is higher than the wt itself. Which concentration of Tu and which time point have been used?

-Fig 1D and E The release of cytochrome C (cytc) should be easily evaluated by WB as well of the mitochondria and cytosolic fractions. How do the authors determine the % of cytc release from the confocal images is this % of cells with loss of mitochondrial cytc? Ctrs conditions should also be shown in Fig. 1E. Can the authors also comment on the cytc difference between MITOL F/F vs MITOL -/- in ctrls.

-Fig S2A is missing the tot level of PERK and the not cleaved form of ATF6 as well as the ratio between the activated form and the total uncleaved form.

-Fig S3A,B: From the representative image it seems that the mitochondria network in MITOL KO is more fragmented after Tu in comparison to MITOL F/F. Is this the case?

-Fig 4B: in the IP the HA detection is stronger in the condition not transfected with IRE1-FLAG. Although the interaction is convincing, perhaps the authors could show a better representative IP.

- Fig. 4C-D; Using GST-pull down assays the authors show that the cytosolic C-terminus of IRE1a interacts with the N-terminus of MITOL. While this analysis is relevant the authors should also show and compare the interaction of the IRE1-C fragment with the full length of IRE1 and the (lack of interaction) with IRE1 luminal domain.

-Fig 5C,D: It would be better to perform same experiment after inducing prolonged ER stress and see if the expression of this mutant can increase IRE1-mediated apoptosis. In basal conditions the role of IRE1a ubiquitilation is less evident.

The legends are often lacking experimental details (for example the concentration of Tu is not always mentioned). Some figure/panel number are not correctly mentioned or labeled (see Fig S3G,S3H: cited in the text, but it's actually S3F and S3G; there are 2 panels labelled as S4C etc).

The English grammar of the manuscript could be improved as well.

Referee #2:

The paper entitled "MITOL prevents ER stress-induced apoptosis by IRE1 α ubiquitylation at mitochondria-ER contact sites" by Shigeru Yanagi and coworkers reported that MITOL inhibits ER stress-induced apoptosis by IRE1 α ubiquitylation at MAM. MITOL conjugates K63 polyubiquitin chain to K481 of IRE1, thereby preventing hyper-oligomerization of IRE1 α and regulated RIDD activity. The mechanism is novel and is important for Ire1 α signaling, which is, in general, supported by the data presented. However, the manuscript was poorly and carelessly written, which makes it extremely hard to understand. It is particularly troubling that discrepancy in descriptions in the text and figures, lack of detailed experimental information, mislabeling of some figures, and grammatic errors were frequently seen throughout the paper. Additional experiments are required to strengthen the conclusion.

Major points:

1. Two shDRP1 were used to detect the knock down efficiency in Figure S1C, but shDRP1#1 was not effective. Figure S1C should be repeated by using at least two effective shDrp1 to avoid off-target. Alternatively, RNAi-rescue experiments should be performed for shDRP#1.
2. According to Figure S1 legend, percentages of cells with abnormal mitochondria or ER were calculated from 100 cells. How the abnormal mitochondria and ER morphologies were defined and how the abnormal cells were counted? The authors need to provide more information.

3. For quantification of Western blots, images and assays, for example, in Figure 1A, B, C, are the averages quantified from independent experiments or experimental replicates? No related information was found in either the figure legends or Methods section.
4. Figure labels are not clear. For example, what's the "ctrl" and "Tu" stand for in Figure 1D and 1E? It's confusing because both of them could not be found in the Western Blotting result (Figure 1D) or imaging graph (Figure 1E). Based on the labels, all cells seemed to be treated with tunicamycin in Figure 1E, but the graph has an untreated control! Similar problem can be found in other figures. Please clarify accordingly.
5. Figure S4C showed that IRE1 α ubiquitylation was reduced by prolonged tunicamycin treatment time from 3 h to 15 h, but the correlation between reduction in IRE1 ubiquitylation and apoptotic response could not be reflected by Figure S4D (Page 12, line 16-20). Figure S4D only showed the importance of MITOL on cell death, it could not be used to demonstrate the relationship between IRE1 ubiquitylation reduction and apoptotic response. What's more, cell death is not equal to apoptosis. Additional experiments are required to prove the conclusion.
6. This article describes MITOL prevents ER stress-induced apoptosis by IRE1 ubiquitylation at MAM. But Figure 6B showed only cofractionation of MITOL and Ire1 in MAM and did not show regulation of Ire1 localization to MAM by MITOL. Therefore, the same experiment should be performed in MITOL KO cells and tunicamycin-treated MEF cells.
7. For Figure 3B, the description in the result section (Page 10, line 12-13) said the endogenous Ire1 was examined, but the Figure legend (Page 39, line 10-11) described the use of transfected Ire1 α -GFP. Which one is true?? If the experiment is based on transfected cells, endogenous Ire1 needs to be examined, because GFP tag may have unanticipated effects on the results due to overexpression and GFP tag.
8. Why the acute MITOL depletion by 4-OHT treatment caused morphology abnormalities both in the ER and mitochondria while MEFs under chronic MITOL knockout (KO) only showed abnormal morphology in ER (Figure S1)?
9. In Figure S4E, the left two lanes were marked the same (MITOL-HA alone) but present a totally different results (ubiquitylation in the second lane), what does that mean? If the first lane is control, the author should correct the label for the first lane. Figure S4F has the same problem.
10. According to the text and legend for Figure S4G, the last lane represents the cells were treated with APY29 alone. But the label in graph means the cells were treated with the combination of KIRA6 and APY29. Which one is correct?

Additional points

11. Page 53, line 5: what's the double MITOLF/F (MITOLF/F) mean?
12. Labeling of graphs in several Figures are confusing.
The text labels in Figure S4 are mismarked from graph C to H.
The unit for treatment time in Figure S4D is missing.
The label in Figure S5A is mismatched.
Page 10, line 21: "Figure S3G, S3H" need to be replaced by "Figure S3F, S3G"
13. The chemical or protein names in the overall text should be consistent with that used in Figures. For example, the authors use "APY29" in Figure S4G and materials, while the results description was "APY-29".
14. Page 22, line 3. "Anti-MITOL rabbit polyclonal antibodies was produced as described previously." Please cite the reference.
15. What does "Error bars indicate {plus minus} SD (n=3)" mean (Figure legends)?
16. Page 58, line 10-11. "Protein and mRNA expression levels of MITOL in the spinal cord were confirmed by qRT-PCR (B) and immunoblotting (C)."
17. Many more:
Page 5, line 11: "thorough" need to be replaced by "through"
Page 13, line 9: "KIRA" need to be replaced by "KIRA6"
Page 22, line 9: "was" needs to be replaced by "were"
Page 22, line 12: "cycloheximide" need to be replaced by "Cycloheximide"
Page 22, line 18: "was" needs to be replaced by "were"

Page 22, line 7: "anti-DLP1" need to be replaced by "anti-DRP1"

Page 28, line 9: "Mice" need to be replaced by "mice"

Page 57, line 7: "APY29(2 μ M)" need to be replaced by "APY29(2 μ M)"

Page 41, line 20: "by" need to be replaced by "to"

Page15, line 19; Page28, line 14; Page37, line 4-5; Page50, Figure S6C; Page56, line 10; "ml" needs to be replaced by "mL"

Page 6, line 18-20: the sentence "We first used stable MITOL-KO MEFs to investigate the effects of three ER stress inducers, thapsigargin, tunicamycin, and brefeldin A." is not a completely sentence. Many more are not included in this list.

Referee #3:

Takeda and colleagues suggested a novel mechanism for regulation of IRE1 α -dependent decay of mRNA (RIDD) activity. The authors hypothesized that mitochondrial ubiquitin ligase (MITOL/MARCH5) inhibits RIDD activity by IRE1 α ubiquitylation mitochondria-associated ER membrane (MAM). Whilst I am not an expert in cell biology, it would seem that the experiments were carefully done, presented well and support this hypothesis. However, the authors need to clarify several points:

1. The authors need to consider the possibility that overexpression of IRE1 α and/or its K481R mutant could interfere with hyper-oligomerization of IRE1 α , RIDD, and cell fate.
2. As the K121Y and D123P mutations disturb the initial step of IRE1 activation (dimerization of the Ire1 luminal domain, LD), apparently MITOL should preferentially interact with monomeric Ire1, non-phosphorylated. The author may want to discuss this observation in more details.
3. The authors observed that 4m8C treatment enhanced the MITOL-dependent IRE1 ubiquitination, suggesting that "MITOL preferentially ubiquitinates IRE1 α under conditions that IRE1 α is unrelated to RNase activation" (p. 13). However, it has been demonstrated 4m8C covalently links to Ire1-CD, modifying Ire1 K599, a phosphate-coordinating residue in the kinase domain, and K907, located in the RNAase active site (PNAS, 2012, 109, E869). Consequently, it's unclear whether modification of the active site of the Ire1 kinase domain or the RNAase site is responsible for the observed effect.
4. The author may want to discuss a plausible mechanism by which ubiquitination of K481 affects Ire1 oligomerization. This residue is located at the juxtamembrane CD region, while this juxtamembrane region has been previously suggested to be important for Ire1 oligomerization (e.g., <https://lens.elifesciences.org/05031/>)

Referee #1:

We greatly appreciate your detailed and constructive comments, which have helped to make our paper stronger. We believe that we could address almost all concerns.

General comments:

1) Authors claim that 4-OHT induced MITOL loss induces 'abnormalities' in the mitochondria and ER network. However, from the staining shown in Figure EV1A and EV1B it is not clear how authors quantify these 'abnormalities'. Since persistence of a 'disturbed' ER morphology (fragmentation? not clear, how do the authors quantify these changes?) is observed either acute or chronic loss of MITOL, this requires a better assessment both at the morphological (e.g. using better markers of tubular and luminal ER like e.g. Sec61b and KDEL) and functional levels. This is important also considering that affecting ER-mito contact sites through MITOL expression may have implication in fundamental ER biological functions (e.g. is the ER-Ca²⁺ steady state level affected?). Moreover, can the authors explain why shDrp1 does not rescue completely the mitochondrial abnormalities?

Response:

We apologize that our explanation was not clear. To define the "abnormalities", representative examples showing abnormal organelle network were shown in Figure EV1A and EV1B. According to your helpful comments, the ER networks were re-stained with mCherry-Sec61β in place of ER-DsRed staining (Fig EV1B, EV1D, EV1G, EV1I, EV5E).

In addition, we measured the resting level of ER-Ca²⁺ by using G-CEPIA1er, an ER-Ca²⁺ sensor, as described in a previous report (Hirabayashi, Kwon et al., 2017), and found a slight accumulation of ER-Ca²⁺ in MITOL-KO MEFs (Figure EV1J). The reason of incomplete rescue of mitochondrial morphology by shDrp1 was considered to be resulted from the transfection efficiency around 70%. We believed that mitochondrial morphology was completely rescued in individual cell transfected with shDrp1.

2) The mechanistic aspects on MITOL-mediated protection on ER stress induced apoptosis would need some improvements and further controls. In general, authors should prove their point that MITOL depletion increases specifically ER stress induced apoptotic, therefore caspase-inhibitable cell death, by showing that MITOL then does not affect non-ER stress triggered apoptosis (e.g. by staurosporine or death receptor ligands). Secondly under severe conditions of ER stress induction mitochondrial apoptosis (by engaging BH3 only proteins like BIM and NOXA and Bax/Bak mediated pore forming activity rather than inducing MPT), is regulated by IRE1a RIDD activity towards miR-17 that represses translation of caspase-2, leading to Bid-mediated and Bax/Bak-induced cytochrome C release. Since MITOL deletion causes a reduction of miR-17 upon Tu treatment, the authors could test the involvement of caspase-2 in the mechanism of MITOL-regulated IRE1a dependent cell death. Moreover, it would be interesting to analyse what happens to the interaction between BIM and IRE1a, and to the status of IRE1a-Ub under conditions of ER stress. Since the oligomerization of IRE1 is induced by ER stress, and lack of MITOL induces a hyper-oligomerization of IRE1a, is the interaction between BIM and IRE1 (especially in MITOL KO) increased after Tu, when IRE1 signaling is pro-apoptotic?

Response:

We are very sorry for not explaining enough about this matter. As previously reported (Xu, Cherok et al., 2016), MITOL deletion causes vulnerability against various stimulations including non-ER stresses. Indeed, a previous work have demonstrated an increased apoptosis with staurosporine treatment via accumulation of Mid49/Drp1 in MITOL-KO cells (Xu et al., 2016).

However, in tunicamycin treated MITOL-KO cells, not only mitochondrial permeability transition pore, an end point of cell death signals, but also IRE1 α hyper-activation, an initial step of ER stress response, were significantly induced. Furthermore, this IRE1 α hyper-activation in MITOL-KO cells was independent on Drp1 (Figure EV3A-E). These results suggest that under ER stress MITOL elicits anti-apoptotic effects by direct regulation of the UPR sensor protein, not only through mitochondrial regulation.

We also evaluated *caspase-2* mRNA as another target of miR-17 and found that MITOL KO enhanced *caspase-2* mRNA as well as *txnip* mRNA under ER stress (Figure 2G). According to your suggestion, we examined the changes of MITOL-mediated IRE1 α ubiquitylation after tunicamycin treatment. Tunicamycin decreased MITOL-mediated IRE1 α ubiquitylation and conversely increased IRE1 α -BIM interaction in a time-dependent manner (Figure EV4C, EV4D). Especially, the IRE1 α -BIM interaction was enhanced 15 hours after tunicamycin treatment when cells underwent mild apoptosis.

3) Related to Fig. 2, what is the effect of MITOL loss on UPR signalling at later time points? This point seems relevant since at the early phase of ER stress p-JNK inhibits cell death instead of promoting it (see paper of Brown et al, J Cell Science 2016). Also JNK inhibitors are known to have different off-target effects. Thus the role of JNK in MITOL-mediated IRE1 signalling should be better validated by a genetic approach.

Response:

This is an important question. Interestingly, MITOL loss extended the period of JNK phosphorylation rather than enhancement (Figure 2H). It is possible that the prolonged JNK activation contributes to induction of apoptosis, rather than cell survival. Therefore, we performed rescue experiments using two siJNK1 to avoid off-target effects. Loss of JNK1 in MITOL-KO cells expectedly rescued tunicamycin-induced apoptosis (Figure EV2E).

4) Although both MITOL and IRE1 α have been shown in other studies to be enriched at the MAMs, the hypothesis that MAMs integrity is required for the MITOL-IRE1 interaction and MITOL-mediated Ub, require a more rigorous analysis. For example Fig 6A; the staining of two proteins located at the ER and mitochondria, even if not necessarily enriched at the MAMs, will reveal a partial co-localization because the two organelles are interconnected. Thus this assay does not really prove the presence of these proteins at the MAMs. In Fig6B: CLX is not enriched in the MAM fraction and the mitochondria are contaminated by Tubulin and IRE1. IRE1 is also not so much present in the ER fraction. Also in Fig6E: The over-expression of MITOL-HA is not the same in the input, and it is actually much less in the PACS siRNA condition, thereby it is difficult to compare the two co-IP and to draw conclusions. Importantly, does the IRE1 α K81R mutant show an impaired localization at the MAMs? And do conditions of ER stress weaken the MITOL-IRE1 interaction at the ER-mitochondria contacts?

Response:

We agree with your concerns about the data to examine the role of MAM in MITOL-IRE1 α association. Certainly, immunostaining data of MITOL/IRE1 α is insufficient to demonstrate a MAM localization of MITOL/IRE1 α . Therefore, we moved this data from main figure to expanded view figure EV6A and changed the sentence explaining about new Figure EV6A as followed.

“As previously reported, tagged MITOL showed a mitochondria-like structure, whereas tagged IRE1 α was observed to form an ER-like structure (Figure EV6A). Since some regions of the ER network are connected to mitochondria, the tagged MITOL was partially co-localized with ER-like structure visualized by the tagged IRE1 α (Figure EV6A). The merged image and line profile showed 15 a partial co-localization of MITOL and IRE1 α (Figure EV6A).”

(Result section, P 15, Line 18 – P 16, Line 1)

Moreover, we demonstrated the MAM localization of IRE1 α and MITOL by using highly purified MAM fraction (Figure 6A). The contaminations of IRE1 α and Tubulin were removed from the mitochondria fraction by highly purification. However, CNX was still detected in the MAM fraction. Since several reports have suggested that CNX also localized at the MAM in some cell lines (Arasaki, Shimizu et al., 2015, Hamasaki, Furuta et al., 2013), we consider that the detection of CNX in the MAM fraction is not resulted from technical issues including contamination of other organelle.

Furthermore, MITOL interacts with IRE1 α in ER membrane co-precipitated with mitochondria (Figure 6C). Since MITOL strongly intracts with inactive and monomeric IRE1 α (Figure EV4I, EV4G), this interaction may be attenuated under ER stress. Actually, MITOL-dependent IRE1 α ubiquitylation was attenuated under ER stress (Figure EV4C). Figure 6E was improved by loading equal amount of MITOL-HA in input. In addition, we demonstrated that K48 R mutant did not affect MAM localization of IRE1 α in Figure EV6C.

Specific remarks:

Is the analysis presented in Fig.1C from the same WB? In the WB shown it seems that the MITOL and B-actin bands differ quite strongly, in comparison to the upper bands (cPARP and cC3); not clear also why the molecular weight of the CS mutant, is higher than the wt itself. Which concentration of Tu and which time point have been used?

Response:

According to your kind comment, Figure 1C was improved. Concentration of Tu and time point were properly described in the legend and the methods as followed.

“All experiments using Tu below were performed at concentration of 0.7 $\mu\text{g}/\text{mL}$.”

(Legend section, P 41, Line 6- 7)

“Tunicamycin (Tu) was dissolve in DMSO and used for cellular experiments at 0.7 $\mu\text{g}/\text{mL}$.”

(Methods section, P 23, Line 15 –16)

Fig 1D and E The release of cytochrome C (cytc) should be easily evaluated by WB as well of the mitochondria and cytosolic fractions. How do the authors determine the % of cytc release from the confocal images is this % of cells with loss of mitochondrial cytc? Ctrs conditions should also be shown in Fig. 1E. Can the authors also comment on the cytc difference between MITOL F/F vs MITOL $^{-/-}$ in ctrls.

Response:

We appreciate your suggestion. To evaluate the release of cytochrome C accurately, subcellular fractionation assay was performed instead of immunostaining data. We could obtain a convincing result showing the release of cytochrome C in tunicamycin-treated MITOL-KO cells (Figure 1D). Immunoblot analysis revealed that there was no change in cytochrome C release between MITOL^{F/F} and MITOL^{-/-} cells in the basal conditions. These difference of results between immunoblot analysis and immunostaining may be due to the lack of accuracy in the quantification of immunostaining data as pointed out by the referee.

Fig EV2A is missing the tot level of PERK and the not cleaved form of ATF6 as well as the ratio between the activated form and the total uncleaved form.

Response:

Following your concerns, total level of PERK was shown by immunoblot analysis in Figure EV2A and uncleaved form of ATF6 was added in Figure EV2B. Since uncleaved ATF6 was not drastically changed in indicated periods compared to cleaved ATF6, we normalized cleaved ATF6 by the amount of β -actin.

Fig EV3A,B: From the representative image it seems that the mitochondria network in MITOL KO is more fragmented after Tu in comparison to MITOL F/F. Is this the case?

Response:

We agree with your comment. We consider that MITOL-KO mitochondria are prone to fragmented under various stress conditions, however, no statistically significant change in mitochondrial morphology between MITOL^{F/F} and MITOL^{-/-} cells was observed at least 4 hours after tunicamycin treatment.

Fig 4B: in the IP the HA detection is stronger in the condition not transfected with IRE1-FLAG. Although the interaction is convincing, perhaps the authors could show a better representative IP.

Response:

Figure 4B was properly improved according to this comment.

Fig. 4C-D; Using GST-pull down assays the authors show that the cytosolic C-terminus of IRE1a interacts with the N-terminus of MITOL. While this analysis is relevant the authors should also show and compare the interaction of the IRE1-C fragment with the full length of IRE1 and the (lack of interaction) with IRE1 luminal domain.

Response:

Your concern is reasonable. Although we challenged to overcome this problem, unfortunately, we failed to generate GST-fused full length of IRE1 α due to the huge protein with a transmembrane domain. We succeed to generate GST-fused luminal domain, but could not obtain enough amount of it due to the extremely low expression in E.coli. However, since C-terminus of IRE1 α is exposed to cytosol, the most likely conclusion is that MITOL interacts with IRE1 α via the domain exposed to cytosol, rather than the luminal domain.

Fig 5C,D: It would be better to perform same experiment after inducing prolonged ER stress and see if the expression of this mutant can increase IRE1-mediated apoptosis. In basal conditions the role of IRE1a ubiquitilation is less evident.

Response:

We thank you for your nice suggestion. We added the results showing enhanced cell death by K481R mutant under tunicamycin stimulation in Figure EV5C and EV5D. IRE1 α K481R strongly induced cell death under ER stress.

The legends are often lacking experimental details (for example the concentration of Tu is not always mentioned). Some figure/panel number are not correctly mentioned or labeled (see Fig EV3G,EV3H: cited in the text, but it's actually EV3F and EV3G; there are 2 panels labelled as EV4C etc).

Response:

We are sorry for these careless mistakes. We corrected these points.

The English grammar of the manuscript could be improved as well.

Response:

The English grammar was checked again and improved by a native speaker.

Referee #2:

First of all, we would like to apologize for many our careless mistakes in the text. We are very grateful to you for pointing out our mistakes.

Major points:

1. Two shDRP1 were used to detect the knock down efficiency in Figure EV1C, but shDRP1#1 was not effective. Figure EV1C should be repeated by using at least two effective shDrp1 to avoid off-target. Alternatively, RNAi-rescue experiments should be performed for shDRP#1.

Response:

We thank you for your suggestion. According to this comment, we re-constructed shDrp1#1 and Figure EV1E, EV1F, EV1G were improved by using two effective shDrp1 to avoid off-target effects.

2. According to Figure EV1 legend, percentages of cells with abnormal mitochondria or ER were calculated from 100 cells. How the abnormal mitochondria and ER morphologies were defined and how the abnormal cells were counted? The authors need to provide more information.

Response:

We apologize that our explanation was not clear. To define the “abnormalities”, representative examples showing abnormal organelle network were shown in Figure EV1A, EV1B. The cells with abnormal organelle network were counted by visual observation and this was described in each legend as followed.

“Percentages of cells with abnormal mitochondria were calculated from 100 cells by visual inspection in each independent experiment.”

(Legend section, P 43 - 44, P 46,)

(Legend section of expanded view figure, P 1 – P3)

3. For quantification of Western blots, images and assays, for example, in Figure 1A, B, C, are the averages quantified from independent experiments or experimental replicates? No related information was found in either the figure legends or Methods section.

Response:

We appreciate your comments. The averages were quantified from independent experiments. We described this in the legends and Methods section as followed.

“Percentages of cells with abnormal mitochondria were calculated from 100 cells by visual inspection in each independent experiment.”

(Legend section, P 43 - 44, P 46,)

(Legend section of expanded view figure, P 1 – P3)

“**Statistical analysis.** All results are expressed as mean \pm SD. Obtained data were compared between independent experiments using either two-tailed Student t-test. The number of independent experiments is shown as n.”

(Method section, P 32, Line 4 – 7)

4. Figure labels are not clear. For example, what's the "ctrl" and "Tu" stand for in Figure 1D and 1E? It's confusing because both of them could not be found in the Western Blotting result (Figure 1D) or imaging graph (Figure 1E). Based on the labels, all cells seemed to be treated with tunicamycin in Figure 1E, but the graph has an untreated control! Similar problem can be found in other figures. Please clarify accordingly.

Response:

We are sorry for unclear labels. Detailed descriptions about Ctrl and Tu were added in the legends and the methods as followed.

“Control MEFs (MITOL^{+/+}) and MITOL-KO MEFs (MITOL^{-/-}) were treated with DMSO as control, 0.8 μM Thapsigargin (Tg), 0.7 μg/mL Tunicamycin (Tu) or 1.2 μg/mL Brefeldin A (Br) for 24 hours.”

(Legend section, P 41, Line 4 – 6)

“Tunicamycin (Tu) was dissolve in DMSO and used for cellular experiments at 0.7 μg/mL. DMSO was also treated to cells as control for Tu.”

(Method section, P 23, Line 15 – 16)

According to comment of other referee, subcellular fractionation assay was performed instead of immunostaining data to evaluate the release of cytochrome C accurately (Figure 1E). We could obtain a convincing result showing the release of cytochrome C in tunicamycin-treated MITOL-KO cells (Figure 1D). This difference of results between immunoblot analysis and immunostaining may be due to the lack of accuracy in the quantification of immunostaining data as pointed out by other referee. Tunicamycin-untreated cells were also quantified from immunoblot data and the result was represented by means of a graph.

5. Figure EV4C showed that IRE1 ubiquitylation was reduced by prolonged tunicamycin treatment time from 3 h to 15 h, but the correlation between reduction in IRE1 ubiquitylation and apoptotic response could not be reflected by Figure EV4D (Page 12, line 16-20). Figure EV4D only showed the importance of MITOL on cell death, it could not be used to demonstrate the relationship between IRE1 ubiquitylation reduction and apoptotic response. What's more, cell death is not equal to apoptosis. Additional experiments are required to prove the conclusion.

Response:

We apologize that our explanation was not proper as pointed out. Prolonged tunicamycin treatment reduces MITOL-dependent IRE1α ubiquitylation, thereby permits IRE1α over-activation, leading to apoptosis. Therefore, it could be predicted that reduction of IRE1α ubiquitylation precedes apoptosis, rather than occurring at the same time. Actually, reduction of IRE1α ubiquitylation preceded apoptosis (Figure EV4E). We thus changed the sentence in the result as follows.

“This reduction in IRE1α ubiquitylation preceded apoptotic response following tunicamycin treatment”

(Result section, P 12, Line 22- P 13, Line 1)

6. This article describes MITOL prevents ER stress-induced apoptosis by IRE1 ubiquitylation at MAM. But Figure 6B showed only cofractionation of MITOL and Ire1 in MAM and did not show regulation of Ire1 localization to MAM by MITOL. Therefore, the same experiment should be performed in MITOL KO cells and tunicamycin-treated MEF cells.

Response:

We agree with your suggestion. To demonstrate that IRE1α is regulated by MITOL at the MAM, immunoprecipitation assay was performed using isolated organelle membrane fractions. We found that MITOL especially interacts with IRE1α in ER membrane co-precipitated with mitochondria (Figure 6C). Furthermore, overexpressing MITOL specifically ubiquitylated MAM-localized IRE1α (Figure 6D), indicating that MITOL regulates IRE1α at the MAM. In addition, we checked that MITOL loss did not affect MAM localization of IRE1α (Figure EV6B).

7. For Figure 3B, the description in the result section (Page 10, line 12-13) said the endogenous Ire1 was examined, but the Figure legend (Page 39, line 10-11) described the use of transfected Ire1^{Δj}-GFP. Which one is true?? If the experiment is based on transfected cells, endogenous Ire1 needs to be examined, because GFP tag may have unanticipated effects on the results due to overexpression and GFP tag.

Response:

We are sorry for our careless mistake. We used endogenous IRE1α in Figure 3B and corrected the mistake as followed.

“To evaluate the oligomerization level of endogenous IRE1α, cells without any transfection were solubilized and separated by sucrose density-gradient centrifugation, followed by immunoblotting with anti-IRE1α antibody (B)”
(Legend section, P 43, Line 15 – 17)

8. Why the acute MITOL depletion by 4-OHT treatment caused morphology abnormalities both in the ER and mitochondria while MEFs under chronic MITOL knockout (KO) only showed abnormal morphology in ER (Figure EV1)?

Response:

We agree with your question. Acute MITOL deletion induces a rapid accumulation of Drp1 that leads to mitochondrial fragmentation (Figure EV1E, EV1F). However, in chronic MITOL-KO cells, Drp1 accumulation was attenuated and mitochondrial morphology was restored to an almost normal level (Figure EV3A), suggesting that a compensatory change occurs to inhibit Drp1. Since abnormality in the ER was observed in both acute and chronic MITOL-KO cells, we hypothesize that a similar compensatory mechanism does not exist in the ER. The detailed mechanism is currently under investigation and will be reported in the near future.

9. In Figure EV4E, the left two lanes were marked the same (MITOL-HA alone) but present a totally different results (ubiquitylation in the second lane), what does that mean? If the first lane is control, the author should correct the label for the first lane. Figure EV4F has the same problem.

Response:

We are sorry for this mistake. As you pointed out, first lane is control. We corrected the mistake.

10. According to the text and legend for Figure EV4G, the last lane represents the cells were treated with APY29 alone. But the label in graph means the cells were treated with the combination of KIRA6 and APY29. Which one is correct?

Response:

Thank you for pointing out this mistake. As you pointed out, last lane is treated with APY-29 only. We corrected this.

Additional points

11. Page 53, line 5: what's the double MITOLF/F (MITOLF/F) mean?

12. Labeling of graphs in several Figures are confusing.

The text labels in Figure EV4 are mismarked from graph C to H.

The unit for treatment time in Figure EV4D is missing.

The label in Figure EV5A is mismatched.

Page 10, line 21: "Figure EV3G, EV3H" need to be replaced by "Figure EV3F, EV3G"
13. The chemical or protein names in the overall text should be consistent with that used in Figures. For example, the authors use "APY29" in Figure EV4G and materials, while the results description was "APY-29".

14. Page 22, line 3. "Anti-MITOL rabbit polyclonal antibodies was produced as described previously." Please cite the reference.

15. What does "Error bars indicate {plus minus} SD (n=3)" mean (Figure legends)?

16. Page 58, line 10-11. "Protein and mRNA expression levels of MITOL in the spinal cord were confirmed by qRT-PCR (B) and immunoblotting (C)."

17. Many more: Page 5, line 11: "thorough" need to be replaced by "through"

Page 13, line 9: "KIRA" need to be replaced by "KIRA6"

Page 22, line 9: "was" needs to be replaced by "were"

Page 22, line 12: "cycloheximide" need to be replaced by "Cycloheximide"

Page 22, line 18: "was" needs to be replaced by "were"

Page 22, line 7: "anti-DLP1" need to be replaced by "anti-DRP1"

Page 28, line 9: "Mice" need to be replaced by "mice"

Page 57, line 7: "APY29(2 μ M)" need to be replaced by "APY29(2 μ M)"

Page 41, line 20: "by" need to be replaced by "to"

Page15, line 19; Page28, line 14; Page37, line 4-5; Page50, Figure EV6C; Page56, line 10; "ml" needs to be replaced by "mL"

Page 6, line 18-20: the sentence "We first used stable MITOL-KO MEFs to investigate the effects of three ER stress inducers, thapsigargin, tunicamycin, and brefeldin A." is not a completely sentence.

Many more are not included in this list.

Response:

We are very sorry for these careless mistakes and thank you for pointing them out. We corrected these mistakes and others carefully.

Referee #3:

We would like to express our appreciation for your comments which could strengthen our paper.

clarify several points:

1. The authors need to consider the possibility that overexpression of IRE1 α and/or its K481R mutant could interfere with hyper-oligomerization of IRE1 α , RIDD, and cell fate.

Response:

Thank you for your suggestion. Since these are several studies using IRE1 α overexpression to overcome regulatory effect of BiP (Ghosh, Wang et al., 2014, Li, Korennykh et al., 2010), we consider that over-activation of IRE1 α by K481R mutation as shown in Figure 5 is not due to the indirect effect. We also challenged to address your concern that K481R mutant-mediated apoptosis is merely caused by overexpression of insoluble proteins, we examined whether IRE1 α K481R mutant is functional protein and induces cell death in an ER stress-dependent manner. We found that IRE1 α K481R mutant exhibited a strong RNase activity and enhanced both excessive oligomer formation and apoptosis in a tunicamycin-dependent manner (Figure 5G, EV5A, EV5C, EV5D). Thus, we conclude that K481R mutation induces apoptosis via enhancement of IRE1 α -specific activation.

2. As the K121Y and D123P mutations disturb the initial step of IRE1 activation (dimerization of the Ire1 luminal domain, LD), apparently MITOL should preferentially interact with monomeric Ire1, non-phosphorylated. The author may want to discuss this observation in more details.

Response:

Following your comment, we changed and added the more detailed explanation in the result section described Figure EV4G as followed.

“Upon ER stress, IRE1 α initially undergoes self-association through its luminal domain, leading to trans-phosphorylation and then IRE1 α interfaces via its cytosolic domain allowing for mRNA docking onto IRE1 α and RNase activation. To understand the molecular mechanism behind the recognition of IRE1 α by MITOL, cells were transfected with K121Y or D123P mutants of IRE1 α , which lack the ability of luminal self-association (Li et al., 2010, Zhou, Liu et al., 2006). IRE1 α ubiquitylation by MITOL was enhanced by the mutation of K121Y or D123P (Figure EV4G), suggesting that MITOL preferentially ubiquitylates monomeric IRE1 α .”

(Result section, P 13, Line 4 – 11)

3. The authors observed that 4m8C treatment enhanced the MITOL-dependent IRE1 ubiquitination, suggesting that "MITOL preferentially ubiquitinates IRE1 α under conditions that IRE1 α is unrelated to RNase activation" (p. 13). However, it has been demonstrated 4m8C covalently links to Ire1-CD, modifying Ire1 K599, a phosphate-coordinating residue in the kinase domain, and K907, located in the RNAase active site (PNAS, 2012, 109, E869). Consequently, it's unclear whether modification of the active site of the Ire1 kinase domain or the RNAase site is responsible for the observed effect.

Response:

We appreciate your kind comment. This is very interesting point. We checked this previous study and changed the explanation of Figure EV4H, EV4I (Result section, P 13, Line 14 – P 14, Line 1). Certainly, 4 μ 8C was reported to interact not only K907 in the RNase domain but also K599 in the kinase domain of IRE1 α (Cross, Bond et al., 2012), although the physiological effect of 4 μ 8C is suggested to be limited to the RNase of IRE1 α due to the competition with endogenous nucleotides for the binding to K907. Since it is possible that

4 μ 8C also inhibits kinase activity of IRE1 α , we used two IRE1 α kinase inhibitors, APY-29 and KIRA6. Interestingly, allosteric modulation of APY-29 leads to IRE1 α oligomerization and RNase activation, whereas, KIRA6 inhibits both oligomerization and RNase activation. Although KIRA6 increased IRE1 α ubiquitylation by MITOL, APY-29 decreased IRE1 α ubiquitylation (Figure EV4J). Thus, a likely conclusion could be that MITOL specifically ubiquitylates RNase-inactive form of IRE1 α including IRE1 α monomer.

4. The author may want to discuss a plausible mechanism by which ubiquitination of K481 affects Ire1 oligomerization. This residue is located at the juxtamembrane CD region, while this juxtamembrane region has been previously suggested to be important for Ire1 oligomerization (e.g., <https://lens.elifesciences.org/05031/>)

Response:

We are grateful for your supportive comment that helped us strengthen our data. We mentioned the functional relationship between IRE1 α K481 and its oligomerization in based on the previous study.

“A recent study has demonstrated that the basic residues in the juxtamembrane region of IRE1 α cytosolic domain contribute to mRNA docking onto its 19 oligomers. Since the basic residue K481 of IRE1 α , a specific site ubiquitylated by MITOL, is located in the juxtamembrane region, MITOL may regulate IRE1 α RNase activity via mRNA docking, in addition to the stability of its oligomers.”

(Discussion section, P 19, Line 21 – P 20, Line 3).

References

- Arasaki K, Shimizu H, Mogari H, Nishida N, Hirota N, Furuno A, Kudo Y, Baba M, Baba N, Cheng J, Fujimoto T, Ishihara N, Ortiz-Sandoval C, Barlow LD, Raturi A, Dohmae N, Wakana Y, Inoue H, Tani K, Dacks JB et al. (2015) A role for the ancient SNARE syntaxin 17 in regulating mitochondrial division. *Dev Cell* 32: 304-17
- Cross BC, Bond PJ, Sadowski PG, Jha BK, Zak J, Goodman JM, Silverman RH, Neubert TA, Baxendale IR, Ron D, Harding HP (2012) The molecular basis for selective inhibition of unconventional mRNA splicing by an IRE1-binding small molecule. *Proc Natl Acad Sci USA* 109: E869-78
- Ghosh R, Wang L, Wang ES, Perera BG, Igbaria A, Morita S, Prado K, Thamsen M, Caswell D, Macias H, Weiberth KF, Gliedt MJ, Alavi MV, Hari SB, Mitra AK, Bhatarai B, Schurer SC, Snapp EL, Gould DB, German MS et al. (2014) Allosteric inhibition of the IRE1alpha RNase preserves cell viability and function during endoplasmic reticulum stress. *Cell* 158: 534-48
- Hamasaki M, Furuta N, Matsuda A, Nezu A, Yamamoto A, Fujita N, Oomori H, Noda T, Haraguchi T, Hiraoka Y, Amano A, Yoshimori T (2013) Autophagosomes form at ER-mitochondria contact sites. *Nature* 495: 389-393
- Hirabayashi Y, Kwon SK, Paek H, Pernice WM, Paul MA, Lee J, Erfani P, Raczkowski A, Petrey DS, Pon LA, Polleux F (2017) ER-mitochondria tethering by PDZD8 regulates Ca(2+) dynamics in mammalian neurons. *Science* 358: 623-630
- Li H, Korennykh AV, Behrman SL, Walter P (2010) Mammalian endoplasmic reticulum stress sensor IRE1 signals by dynamic clustering. *Proc Natl Acad Sci USA* 107: 16113-8
- Xu S, Cherok E, Das S, Li S, Roelofs BA, Ge SX, Polster BM, Boyman L, Lederer WJ, Wang C, Karbowski M (2016) Mitochondrial E3 ubiquitin ligase MARCH5 controls mitochondrial fission and cell sensitivity to stress-induced apoptosis through regulation of MiD49 protein. *Mol Biol Cell* 27: 349-359

2nd Editorial Decision

10th April 2019

Thank you for submitting a revised version of your manuscript. It has now been seen by two of the original referees whose comments are appended below.

As you will see, while referee #2 finds that his/her criticisms have been sufficiently addressed and recommends the manuscript for publication, referee #1 remains concerned about a few minor points. In particular, s/he requests you to describe how ER abnormalities in Fig. EV1A/1B are quantified, as well as to improve the mitochondrial and MAM purification in Fig. 6A and the CoIP experiment in Fig. 4B. Also, control blots for these experiments have to be provided. We agree with referee #1 that these are important points that should be addressed before we can officially accept your manuscript for publication here.

In addition to resolving these concerns from referee #1, there are a few editorial issues about the text and the figures that I need you to address:

REFEREE REPORTS

Referee #1:

The revised manuscript has been largely improved by the authors. However, there are still some minor points that the authors did not address satisfactorily. These are listed below.

Fig. EV1A/1B; the authors here should still define how they quantify these ER abnormalities. Please describe it.

Fig. 6A: Likely authors misunderstood the previous comment/concern of the reviewer, about Fig6B. 'In Fig6B: CLX is not enriched in the MAM fraction'. The point was exactly meant to say that CNX should be enriched at MAM since it is a MAM-protein and during purification of this fraction CNX should be indeed found 'enriched'.

In the new WB blot of Fig6A, CNX is again not enriched as it should in both, MAM and ER fractions, and neither is the mitochondrial protein TOM20 in the corresponding mitochondrial fraction. Hence the mitochondria and MAM purification -also in the absence of the detection of other mitochondrial proteins known to be present at MAMs, such MFN2 and/or VDAC1- still remains to be ameliorated.

In analogy, Fig EV6C, is not only missing an ER marker in the ER+CYT fraction and an MAM marker, but again shows that the mito maker TOM20 is not enriched in the mitochondrial fraction. Can the authors explain these inconsistencies?

Also Fig4B remains only partially improved. It is strange to see that there is basically the same amount of HA both in the pulled down (IP) fraction and in the input.

Referee #2:

My concerns have been satisfactorily addressed in this revised manuscript.

Corresponding Author Name: Shigeru Yanagi

Journal Submitted to: The EMBO Journal

Manuscript Number: EMBOJ-2018-100999R